# SketchEmbedNet: Learning Novel Concepts by Imitating Drawings

## Abstract

Sketch drawings are an intuitive visual domain that appeals to human instinct. Previous work has shown that recurrent neural networks are capable of producing sketch drawings of a single or few classes at a time. In this work we investigate representations developed by training a generative model to produce sketches from pixel images across many classes in a sketch domain. We find that the embeddings learned by this sketching model are extremely informative for visual tasks and infer a unique visual understanding. We then use them to exceed state-of-the-art performance in unsupervised few-shot classification on the Omniglot and mini-ImageNet benchmarks. We also leverage the generative capacity of our model to produce high quality sketches of novel classes based on just a single example.

## 1 Introduction

Upon encountering a novel concept, such as a six-legged turtle, humans can quickly generalize this concept by composing a mental picture. The ability to generate drawings greatly facilitates communicating new ideas. This dates back to the advent of writing, as many ancient written languages are based on logograms, such as Chinese hanzi and Egyptian hieroglyphs, where each character is essentially a sketch of the object it represents. We often see complex visual concepts summarized by a few simple strokes.

Inspired by the human ability to draw, recent research has explored the potential to generate sketches using a wide variety of machine learning models, ranging from hierarchical Bayesian models (Lake et al., 2015), to more recent deep autoregressive models (Gregor et al., 2015; Ha & Eck, 2018; Chen et al., 2017) and generative adversarial nets (GANs) (Li et al., 2019). It is a natural question to ask whether we can obtain useful intermediate representations from models that produce sketches in the output space, as has been shown by other generative models (Ranzato et al., 2006; Kingma & Welling, 2014; Goodfellow et al., 2014; Donahue et al., 2017; Doersch et al., 2015). Unfortunately, a hierarchical Bayesian model suffers from prolonged inference time, while other current sketch models mostly focus on producing drawings in a closed set setting with a few classes (Ha & Eck, 2018; Chen et al., 2017), or on improving log likelihood at the pixel level (Rezende et al., 2016). Leveraging the learned representation from these drawing models remains a rather unexplored topic.

In this paper, we pose the following question: *Can we learn a generalized embedding function that captures salient and compositional features by directly imitating human sketches?* The answer is affirmative. In our experiments we develop SketchEmbedNet, an RNN-based sketch model trained to map grayscale and natural image pixels to the sketch domain. It is trained on hundreds of classes without the use of class labels to learn a robust drawing model that can sketch diverse and unseen inputs. We demonstrate salience by achieving state-of-the-art performance on the Omniglot few-shot classification benchmark and visual recognizability in one-shot generations. Then we explore how the embeddings capture image components and their spatial relationships to explore image space compositionality and also show a surprising property of conceptual composition.

We then push the boundary further by applying our sketch model to natural images—to our knowledge, we are the first to extend stroke-based autoregressive models to produce drawings of open domain natural images. We train our model with adapted SVG images from the Sketchy dataset (Sangkloy et al., 2016) and then evaluate the embedding quality directly on unseen classes in the mini-ImageNet task for few-shot classification (Vinyals et al., 2016). Our approach is competitive with existing unsupervised few-shot learning methods (Hsu et al., 2019; Khodadadeh et al., 2019; Antoniou & Storkey, 2019) on this natural image benchmark. In both the sketch and natural image domain, we show that by learning to draw, our methods generalize well even across different datasets and classes.

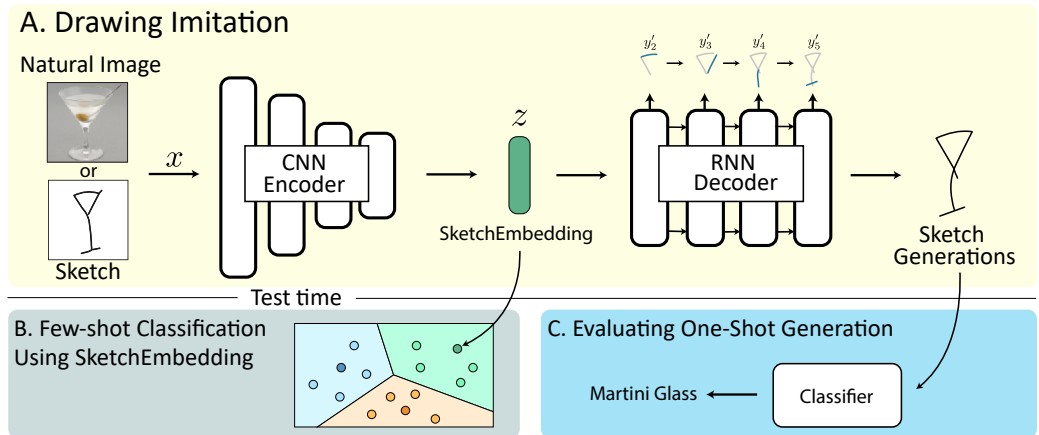

Figure 1: **A:** A natural or sketch pixel image is passed into the CNN encoder to obtain Gaussian SketchEmbedding $z$, which is concatenated with the previous stroke $y_{t-1}$ as the decoder input at each timestep to generate $y_t$. **B+C:** Downstream tasks performed after training is complete.

## 2 RELATED WORK

In this section we review relevant literature including generating sketch-like images, unsupervised representation learning, unsupervised few-shot classification and sketch-based image retrieval (SBIR).

**Autoregressive drawing models:** Graves (2013) use an LSTM to directly output the pen coordinates to imitate handwriting sequences. SketchRNN (Ha & Eck, 2018) builds on this by applying it to general sketches beyond characters. Song et al. (2018); Cao et al. (2019); Ribeiro et al. (2020) all extend SketchRNN through architectural improvements. Chen et al. (2017) change inputs to be pixel images. This and the previous 3 works consider multi-class sketching, but none handle more than 20 classes. Autoregressive models also generate images directly in the pixel domain. DRAW (Gregor et al., 2015) uses recurrent attention to plot pixels; Rezende et al. (2016) extends this to one-shot generation and PixelCNN (van den Oord et al., 2016) generates image pixels sequentially.

**Image processing methods & GANs:** Other works produce sketch-like images based on style transfer or low-level image processing techniques. Classic methods are based on edge detection and image segmentation (Arbelaez et al., 2011; Xie & Tu, 2017). Zhang et al. (2015) use a CNN to directly produce sketch-like pixels for face images. Photo-sketch and pix2pix (Li et al., 2019; Isola et al., 2017) propose a conditional GAN to generate images across different style domains. Image processing based methods do not acquire high-level image understanding, as all the operations are in terms of low-level filtering; none of the GAN sketching methods are designed to mimic human drawings on open domain natural images.

**Unsupervised representation learning:** In the sketch image domain, our method is similar to the large category of generative models which learn unsupervised representations by the principle of analysis-by-synthesis. Work by Hinton & Nair (2005) operates in a sketch domain and learns to draw by synthesizing an interpretable motor program. Bayesian Program Learning (Lake et al., 2015) draws through exact inference of common strokes but learning and inference are computationally challenging. As such, a variety of deep generative models aim to perform approximate Bayesian inference by using an encoder structure that directly predicts the embedding, e.g., deep autoencoders (Vincent et al., 2010), Helmholtz Machine (Dayan et al., 1995), variational autoencoder (VAE) (Kingma & Welling, 2014), BiGAN (Donahue et al., 2017), etc. Our method is also related to the literature of self-supervised representation learning (Doersch et al., 2015; Noroozi & Favaro, 2016; Gidaris et al., 2018; Zhang et al., 2016), as sketch strokes are part of the input data itself. In few-shot learning (Vinyals et al., 2016; Snell et al., 2017; Finn et al., 2017), recent work has explored unsupervised meta-training. CACTUs, AAL and UMTRA (Hsu et al., 2019; Antoniou & Storkey, 2019; Khodadadeh et al., 2019) all operate by generating pseudo-labels for training.

**Sketch-based image retrieval (SBIR):** In SBIR, a model is provided a sketch-drawing and retrieves a photo of the same class. The area is split into fine-grained (FG-SBIR) (Yu et al., 2016; Sangkloy et al., 2016; Bhunia et al., 2020) and a zero-shot setting (ZS-SBIR) (Dutta & Akata, 2019; Pandey et al., 2020; Dey et al., 2019). FG-SBIR considers minute details while ZS-SBIR learns high-level cross-domain semantics and a joint latent space to perform retrieval.

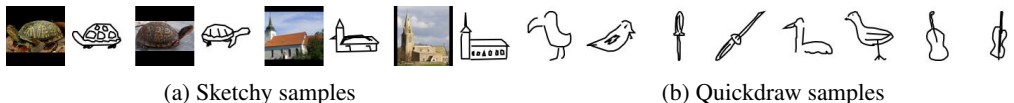

(a) Sketchy samples                           (b) Quickdraw samples

Figure 2: Examples from Sketchy (Sangkloy et al., 2016) and Quickdraw (Jongejan et al., 2016) datasets. Sketchy examples are reshaped and padded to increase image–sketch spatial agreement.

## 3 LEARNING TO IMITATE DRAWINGS

Here we describe learning to draw through sketch imitation. Our architecture is a generative encoder-decoder model with a CNN encoder for pixel images and an RNN decoder to output vector sketches as shown in Figure 1. Unlike other drawing models that only train on a single or few classes (Ha & Eck, 2018; Chen et al., 2017), SketchEmbedNet is not limited by class inputs and can sketch a wide variety of images. We also introduce a differentiable rasterization function for computing an additional pixel-based training loss.

**Input & output representation**   Unlike SketchRNN which encodes drawing sequences, we learn an image embedding by mapping pixels to sketches, similar to Chen et al. (2017). Training data for this task (adopted from Ha & Eck (2018)) consists of a tuple $(\boldsymbol{x}, \boldsymbol{y})$, where $\boldsymbol{x} \in \mathbb{R}^{H \times W \times C}$ is the input image and $\boldsymbol{y} \in \mathbb{R}^{T \times 5}$ is the stroke target. $T$ is the maximum sequence length of the stroke data $\boldsymbol{y}$, and each stroke $\boldsymbol{y}_t$ consists of 5 elements, $(\Delta_x, \Delta_y, s_1, s_2, s_3)$. The first 2 elements are horizontal and vertical displacements on the drawing canvas from the endpoint of the previous stroke. The latter 3 elements are mutually exclusive pen states: $s_1$ indicates the pen is on paper for the next stroke, $s_2$ indicates the pen is lifted, and $s_3$ indicates the sketch sequence has ended. $\boldsymbol{y}_0$ is initialized with (0, 0, 1, 0, 0) to start the generative process. Note that no class information is available while training.

**SketchEmbedding as a compositional encoding of images**   We use a CNN to encode the input image $\boldsymbol{x}$ and obtain the latent space representation $\boldsymbol{z}$, as shown in Figure 1. To model intra-class variance, $\boldsymbol{z}$ is a Gaussian random variable parameterized by CNN outputs, similar to a VAE (Kingma & Welling, 2014). Throughout this paper, we refer to $\boldsymbol{z}$ as the *SketchEmbedding*. In typical image representations the embedding is trained to classify object classes, or to reconstruct the input pixels. Here, since the SketchEmbedding is fed into an RNN decoder to produce a sequence of drawing actions, $\boldsymbol{z}$ is additionally encouraged to have a compositional understanding of the object structure, instead of just an unstructured set of pixel features. For example when drawing the legs of a turtle, the model must explicitly generate each leg instance. While pixel-based models suffer from blurriness and in generating the image at once, does not distinguish between individual components such as the legs, body and head. The loss of this component information by pixel models has been observed in GAN literature (Goodfellow, 2017) which we propose is avoided by our sketching task.

To accommodate the increased training data complexity by including hundreds of classes, we also upscale the size of our model in comparison to work by Chen et al. (2017); Ha & Eck (2018); Song et al. (2018). The backbone is either a 4-layer CNN (Conv4) (Vinyals et al., 2016) for consistent comparisons in the few-shot setting or a ResNet12 (Oreshkin et al., 2018) which produces better drawing results. In comparison, Chen et al. (2017) only use 2D convolution with a maximum of 8 filters.

**RNN decoder**   The RNN decoder used in SketchEmbedNet is the same as in SketchRNN (Ha & Eck, 2018). The decoder outputs a mixture density which represents the stroke distribution at each timestep. Specifically, the stroke distribution is a mixture of some hyperparameter $M$ bivariate Gaussians denoting spatial offsets as well as the probability of the three pen states $s_{1-3}$. The spatial offsets $\boldsymbol{\Delta} = (\Delta x, \Delta y)$ are sampled from the mixture of Gaussians, described by: (1) the normalized mixture weight $\pi_j$; (2) mixture means $\boldsymbol{\mu}_j = (\mu_x, \mu_y)_j$; and (3) covariance matrices $\Sigma_j$. We further reparameterize each $\Sigma_j$ with its standard deviation $\boldsymbol{\sigma}_j = (\sigma_x, \sigma_y)_j$ and correlation coefficient $\rho_{xy,j}$. Thus, the stroke offset distribution is $p(\boldsymbol{\Delta}) = \sum_{j=1}^{M} \pi_j \mathcal{N}(\boldsymbol{\Delta} | \boldsymbol{\mu}_j, \Sigma_j)$.

The RNN is implemented using a HyperLSTM (Ha et al., 2017); LSTM weights are generated at each timestep by a smaller recurrent "hypernetwork" to improve training stability. Generation is autoregressive, using $\boldsymbol{z} \in \mathbb{R}^D$, concatenated with the stroke from the previous timestep $\boldsymbol{y}_{t-1}$, to form the input to the LSTM. Stroke $\boldsymbol{y}_{t-1}$ is the ground truth supervision at train time (teacher forcing), or a sample $\boldsymbol{y}'_{t-1}$, from the mixture distribution output by the model during from timestep $t-1$.

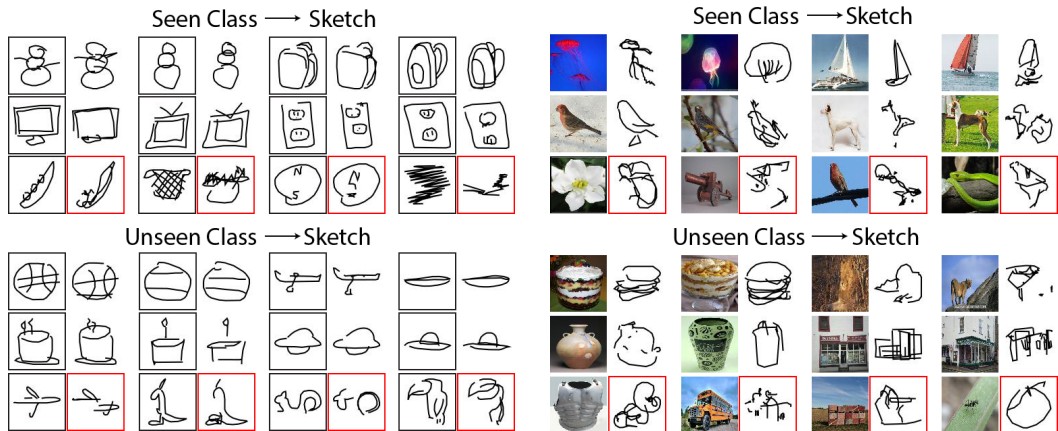

Figure 3: Sampled generated drawings of Quick-draw examples. Weaker drawings boxed in red.

Figure 4: Sampled drawings of mini-ImageNet examples. Weaker drawings boxed in red.

## 3.1 LEARNING

We train the drawing model in an end-to-end fashion by jointly optimizing three losses: a pen loss $\mathcal{L}_{\text{pen}}$ for learning pen states, a stroke loss $\mathcal{L}_{\text{stroke}}$ for learning pen offsets, and our proposed pixel loss $\mathcal{L}_{\text{pixel}}$ for matching the visual similarity of the predicted and the target sketch:

$$\mathcal{L} = \mathcal{L}_{\text{pen}} + (1 - \alpha)\mathcal{L}_{\text{stroke}} + \alpha\mathcal{L}_{\text{pixel}}, \tag{1}$$

where $\alpha$ is a loss weighting hyperparameter. Both $\mathcal{L}_{\text{pen}}$ and $\mathcal{L}_{\text{stroke}}$ were in SketchRNN, while the $\mathcal{L}_{\text{pixel}}$ is our novel contribution to stroke-based generative models. Unlike SketchRNN, we do not impose a prior using KL divergence as we are not interested in unconditional sampling and it worsens quantitative results in later sections.

**Pen loss** The pen-states predictions $\{s'_1, s'_2, s'_3\}$ are optimized as a simple 3-way classification with the softmax cross-entropy loss, $\mathcal{L}_{\text{pen}} = -\frac{1}{T}\sum_{t=1}^{T}\sum_{m=1}^{3} s_{m,t}\log(s'_{m,t})$.

**Stroke loss** The stroke loss maximizes the log-likelihood of the spatial offsets of each ground truth stroke $\mathbf{\Delta}_t$ given the mixture density distribution $p_t$ at each timestep: $\mathcal{L}_{\text{stroke}} = -\frac{1}{T}\sum_{t=1}^{T}\log p_t(\mathbf{\Delta}_t)$.

**Pixel loss** While pixel-level reconstruction objectives are common in generative models (Kingma & Welling, 2014; Vincent et al., 2010; Gregor et al., 2015), we introduce a pixel-based objective for vector sketch generation. After decoding, a *differentiable rasterization* function $f_{\text{raster}}$ is used to map the sketch into a pixel image. $f_{\text{raster}}$ transforms a stroke sequence $\boldsymbol{y}$ into a set of 2D line segments $(l_0, l_1), (l_1, l_2) \ldots (l_{T-1}, l_T)$ where $l_t = \sum_{\tau=0}^{t} \mathbf{\Delta}_\tau$. It renders by fixing canvas dimensions, scaling and centering strokes before determining pixel intensity based on the $L_2$ distance between each pixel to lines in the drawing. Further details on $f_{\text{raster}}$ can be found in Appendix A. $f_{\text{raster}}$ is applied to both the prediction $\boldsymbol{y}'$ and ground truth $\boldsymbol{y}$, to produce two pixel images. Gaussian blur $g_{\text{blur}}(\cdot)$ is used to reduce strictness before computing the binary cross-entropy loss, $\mathcal{L}_{\text{pixel}}$:

$$I = g_{\text{blur}}(f_{\text{raster}}(\boldsymbol{y})), \quad I' = g_{\text{blur}}(f_{\text{raster}}(\boldsymbol{y}')), \quad \mathcal{L}_{\text{pixel}} = -\frac{1}{HW}\sum_{i=1}^{H}\sum_{j=1}^{W} I_{ij}\log(I'_{ij}). \tag{2}$$

**Curriculum training schedule** We find that $\alpha$ (in Equation 1) is an important hyperparameter that impacts both the learned embedding space and the generation quality of SketchEmbedNet. A curriculum training schedule is used, increasing $\alpha$ to prioritize $\mathcal{L}_{\text{pixel}}$ relative to $\mathcal{L}_{\text{stroke}}$ as training progresses; this makes intuitive sense as a single drawing can be produced by many different stroke sequences but learning to draw in a fixed manner is easier. While $\mathcal{L}_{\text{pen}}$ promotes reproducing a specific drawing sequence, $\mathcal{L}_{\text{pixel}}$ only requires that the generated drawing visually matches the image. Like a human, the model should learn to follow one drawing style (a la paint-by-numbers) before learning to draw freely.

## 4 DRAWING IMITATION EXPERIMENTS

In this section, we introduce our experiments on training SketchEmbedNet using two sketching datasets. The first is based on pure stroke-based drawings, and the second consists of natural image and drawing pairs.

Table 1: Few-shot classification results on Omniglot

| Omniglot | | | (way, shot) | | | |
|---|---|---|---|---|---|---|
| Algorithm | Encoder | Train Data | (5,1) | (5,5) | (20,1) | (20,5) |
| Training from Scratch (Hsu et al., 2019) | N/A | Omniglot | 52.50 ± 0.84 | 74.78 ± 0.69 | 24.91 ± 0.33 | 47.62 ± 0.44 |
| CACTUs-MAML (Hsu et al., 2019) | Conv4 | Omniglot | 68.84 ± 0.80 | 87.78 ± 0.50 | 48.09 ± 0.41 | 73.36 ± 0.34 |
| CACTUs-ProtoNet (Hsu et al., 2019) | Conv4 | Omniglot | 68.12 ± 0.84 | 83.58 ± 0.61 | 47.75 ± 0.43 | 66.27 ± 0.37 |
| AAL-ProtoNet (Antoniou & Storkey, 2019) | Conv4 | Omniglot | 84.66 ± 0.70 | 88.41 ± 0.27 | 68.79 ± 1.03 | 74.05 ± 0.46 |
| AAL-MAML (Antoniou & Storkey, 2019) | Conv4 | Omniglot | 88.40 ± 0.75 | 98.00 ± 0.32 | 70.20 ± 0.86 | 88.30 ± 1.22 |
| UMTRA (Khodadadeh et al., 2019) | Conv4 | Omniglot | 83.80 | 95.43 | 74.25 | 92.12 |
| Random CNN | Conv4 | N/A | 67.96 ± 0.44 | 83.85 ± 0.31 | 44.39 ± 0.23 | 60.87 ± 0.22 |
| Conv-VAE | Conv4 | Omniglot | 77.83 ± 0.41 | 92.91 ± 0.19 | 62.59 ± 0.24 | 84.01 ± 0.15 |
| Conv-VAE | Conv4 | Quickdraw | 81.49 ± 0.39 | 94.09 ± 0.17 | 66.24 ± 0.23 | 86.02 ± 0.14 |
| SketchEmbedding (*Ours*) | Conv4 | Omniglot* | 94.88 ± 0.22 | 99.01 ± 0.08 | 86.18 ± 0.18 | 96.69 ± 0.07 |
| SketchEmbedding (*Ours*) | Conv4 | Quickdraw* | **96.96** ± 0.17 | **99.50** ± 0.06 | **91.67** ± 0.14 | **98.30** ± 0.05 |
| MAML (*Supervised*) (Finn et al., 2017) | Conv4 | Omniglot | 94.46 ± 0.35 | 98.83 ± 0.12 | 84.60 ± 0.32 | 96.29 ± 0.13 |
| ProtoNet (*Supervised*) (Snell et al., 2017) | Conv4 | Omniglot | 98.35 ± 0.22 | 99.58 ± 0.09 | 95.31 ± 0.18 | 98.81 ± 0.07 |

*  Stroke data used for training

**Quickdraw: Stroke-based image sketching**    The Quickdraw (Jongejan et al., 2016) dataset consists of 345 classes of each with 70,000 examples, produced by human players participating in the game "Quick, Draw!". Examples from the Quickdraw dataset are shown in Figure 2b. The input image $x$ is a direct rasterization of the drawing data $y$. 300 of 345 classes are randomly selected for training; $x$ is rasterized to a resolution of $28 \times 28$ and stroke labels $y$ padded up to length $T = 64$. Any drawing samples exceeding this length were discarded. Note that this an unsupervised representation learning approach, as no class information is used by the system. Data processing procedures and class splits are in Appendix G.

**Sketchy: Open domain natural image sketching**    We further extend our stroke-based generation model on open domain natural images. Here, the input is an RGB photo, and the output is a human drawing which does not align with the photo precisely and also does not match with the low-level image details. This is a novel setting, as prior efforts by Ha & Eck (2018); Chen et al. (2017); Song et al. (2018) have only applied their sketch RNN models on the Quickdraw dataset or natural images with only two object classes (shoe/chair) and scrubbed backgrounds (Yu et al., 2016). Learning to sketch open domain natural images is very challenging as it requires the model to identify the subject and filter unnecessary details not present in the sketch. At test time, we further challenge our method by evaluating on unseen data distributions necessitating generalization over natural images.

For this task we use the Sketchy dataset (Sangkloy et al., 2016) which consists of ImageNet images paired with vector sketches for a total of 56k examples after processing. Sketches are stored as SVGs with timestamps preserving their original drawing sequence which we adapt by sampling paths in this order. Images are also centered, padded and resized to resolution $84 \times 84$ (see Figure 2a). We fix the maximum sequence length to $T = 100$, and use all 125 categories but remove classes that have overlapping child synsets with the test classes of mini-ImageNet (Vinyals et al., 2016). This enables testing on mini-ImageNet without any alterations to the benchmark. Once again this is an unsupervised learning formulation.

### 4.1    RESULTS AND VISUALIZATIONS

Figure 3 shows drawings conditioned on sketch image inputs. There is little noticeable drop in quality when we sample sketches from unseen classes compared to those it has seen before. Figure 4 shows examples of sketches generated from natural images. While they are not fine-grained renditions, these sketches clearly demonstrate SketchEmbedNet's ability to capture key components of seen and unseen classes. The model effectively isolates the subject in each natural image and captures the circular and square shapes in the cakes and storefronts respectively. Even with the challenging lion images, it identifies the silhouette of the laying lion despite low contrast and appropriately localizes the one on the mountainside.

Unlike pixel-based auto-encoder models, our sketches do not follow the exact pose of the original strokes, but rather capture a general notion of component groupings. In the basketball example of Figure 3, the lines are not a good pixel-wise match to those in the original image yet they are placed in sensible relative positions. Weaker examples are presented in the last row of Figure 3 and 4; regardless, even poorer examples still capture some structural aspects of the original image. Implementation details can be found in Appendix B.

In later sections we explore the uses of SketchEmbeddings and fix models for all downstream tasks.

Table 2: Few-shot classification results on mini-ImageNet

| mini-ImageNet | | | (way, shot) | | | |
|---|---|---|---|---|---|---|
| Algorithm | Backbone | Train Data | (5,1) | (5,5) | (5,20) | (5,50) |
| Training from Scratch (Hsu et al., 2019) | N/A | mini-ImageNet | $27.59 \pm 0.59$ | $38.48 \pm 0.66$ | $51.53 \pm 0.72$ | $59.63 \pm 0.74$ |
| CACTUs-MAML (Hsu et al., 2019) | Conv4 | mini-ImageNet | $39.90 \pm 0.74$ | $53.97 \pm 0.70$ | $63.84 \pm 0.70$ | $69.64 \pm 0.63$ |
| CACTUs-ProtoNet (Hsu et al., 2019) | Conv4 | mini-ImageNet | $39.18 \pm 0.71$ | $53.36 \pm 0.70$ | $61.54 \pm 0.68$ | $63.55 \pm 0.64$ |
| AAL-ProtoNet (Antoniou & Storkey, 2019) | Conv4 | mini-ImageNet | $37.67 \pm 0.39$ | $40.29 \pm 0.68$ | - | - |
| AAL-MAML (Antoniou & Storkey, 2019) | Conv4 | mini-ImageNet | $34.57 \pm 0.74$ | $49.18 \pm 0.47$ | - | - |
| UMTRA (Khodadadeh et al., 2019) | Conv4 | mini-ImageNet | 39.93 | 50.73 | 61.11 | 67.15 |
| Random CNN | Conv4 | N/A | $26.85 \pm 0.31$ | $33.37 \pm 0.32$ | $38.51 \pm 0.28$ | $41.41 \pm 0.28$ |
| Conv-VAE | Conv4 | mini-ImageNet | $23.30 \pm 0.21$ | $26.22 \pm 0.20$ | $29.93 \pm 0.21$ | $32.57 \pm 0.20$ |
| Conv-VAE | Conv4 | Sketchy | $23.27 \pm 0.18$ | $26.28 \pm 0.19$ | $30.41 \pm 0.19$ | $33.97 \pm 0.19$ |
| Random CNN | ResNet12 | N/A | $28.59 \pm 0.34$ | $35.91 \pm 0.34$ | $41.31 \pm 0.33$ | $44.07 \pm 0.31$ |
| Conv-VAE | ResNet12 | mini-ImageNet | $23.82 \pm 0.23$ | $28.16 \pm 0.25$ | $33.64 \pm 0.27$ | $37.81 \pm 0.27$ |
| Conv-VAE | ResNet12 | Sketchy | $24.61 \pm 0.23$ | $28.85 \pm 0.23$ | $35.72 \pm 0.27$ | $40.44 \pm 0.28$ |
| SketchEmbeddingt *(ours)* | Conv4 | Sketchy* | $38.61 \pm 0.42$ | $53.82 \pm 0.41$ | $63.34 \pm 0.35$ | $67.22 \pm 0.32$ |
| SketchEmbedding *(ours)* | ResNet12 | Sketchy* | $\mathbf{40.39} \pm 0.44$ | $\mathbf{57.15} \pm 0.38$ | $\mathbf{67.60} \pm 0.33$ | $\mathbf{71.99} \pm 0.3$ |
| MAML *(supervised)* (Finn et al., 2017) | Conv4 | mini-ImageNet | $46.81 \pm 0.77$ | $62.13 \pm 0.72$ | $71.03 \pm 0.69$ | $75.54 \pm 0.62$ |
| ProtoNet *(supervised)* (Snell et al., 2017) | Conv4 | mini-ImageNet | $46.56 \pm 0.76$ | $62.29 \pm 0.71$ | $70.05 \pm 0.65$ | $72.04 \pm 0.60$ |

* Stroke data used for training

## 5 FEW-SHOT CLASSIFICATION USING SKETCHEMBEDDING

We would like to assess the benefits of learning to draw by performing few-shot classification with our learned embedding space. Examining performance on discriminative tasks reveals that learning to imitate sketches allows the embeddings to capture salient information of novel object classes. Below we describe our few-shot classification procedure and summarize results on the Omniglot (Lake et al., 2015) and mini-ImageNet benchmarks (Vinyals et al., 2016).

**Comparison to unsupervised few-shot classification** In unsupervised few-shot classification, a model is not provided with any class labels during meta-training, until it is given a few labeled examples ("shots") of the novel classes at meta-test time. While our model is provided a "target"—a sequence of strokes—during training, it is not given any class information. Therefore, we propose that the presented sketch imitation training, though it uses sketch sequences, is comparable to other class-label-free representation learning approaches (Berthelot et al., 2019; Donahue et al., 2017; Caron et al., 2018) and the learned SketchEmbeddings can be applied to unsupervised few-shot classification methods.

In our experiments, we compare to previous unsupervised few-shot learning approaches: CAC-TUs (Hsu et al., 2019), AAL (Antoniou & Storkey, 2019), and UMTRA (Khodadadeh et al., 2019). These methods create pseudo-labels during meta-training using either clustering or data augmentation. As additional baselines, a Conv-VAE (Kingma & Welling, 2014) and a random CNN are also included, both using the same Conv4 backbone.

**Few-shot experimental setup** The CNN encoder of SketchEmbedNet is used as an embedding function combined with a linear classification head to perform few-shot classification. The embedding is made deterministic by taking the mean of the random normal latent space $z$ and discarding the variance parameter from the encoder. Otherwise, the conventional episodic setup for few-shot classification is used; each episode consists of a labeled "support" set of $N \times K$ (N-way K-shot) embeddings and an unlabeled "query" set. The linear classification head is trained on the labeled support set and evaluated on the query set.

### 5.1 FEW-SHOT CLASSIFICATION ON OMNIGLOT

The Omniglot (Lake et al., 2015) dataset contains 50 alphabets, 1623 unique character types, each with 20 examples and is presented as both a greyscale image and a stroke drawing. We use the same train-test split as Vinyals et al. (2016) along with randomly sampled episodes. Experiments using the more challenging Lake split where episodes are sampled within alphabet, as proposed by Lake et al. (2015), are in Appendix E and random seed experiments in Appendix F.

To ensure a fair comparison with other few-shot classification models, we use the same convolutional encoder (Conv4) as Vinyals et al. (2016). Results from training only on Omniglot (Lake et al., 2015) are also presented to demonstrate effectiveness without the larger Quickdraw(Jongejan et al., 2016) dataset. No significant improvements were observed using the deeper ResNet12(Oreshkin et al., 2018) architecture; additional results are in Appendix I.

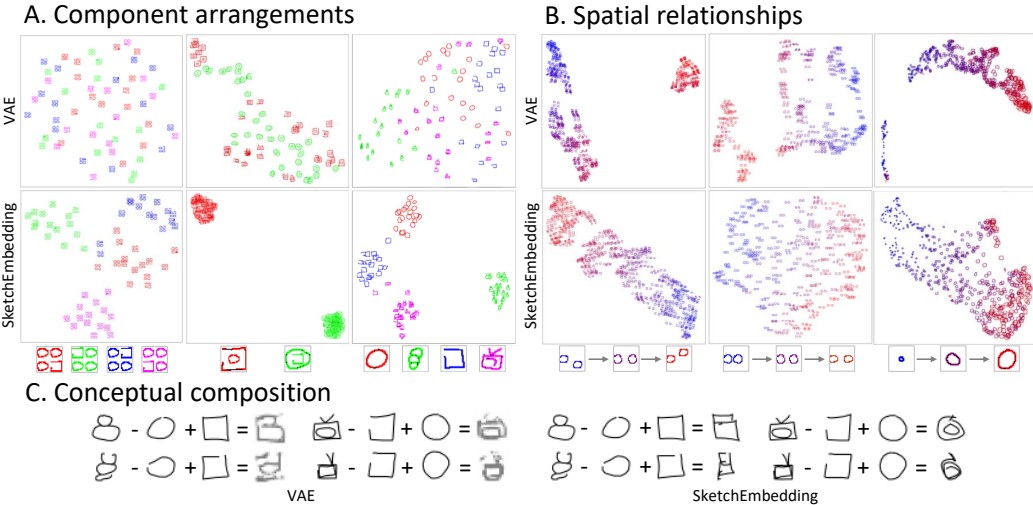

Figure 5: Experiments exploring properties of SketchEmbeddings. Examples colored for understand-ability only.

All of our methods out-perform the previous state-of-the-art on the unsupervised Omniglot benchmark (Table 1). The Quickdraw trained model surpasses supervised MAML (Finn et al., 2017), and is on par with a supervised ProtoNet (Snell et al., 2017) model , especially in the 5-shot settings. Both baselines, a Conv-VAE and a random CNN, perform well compared to other unsupervised methods.

## 5.2 FEW-SHOT CLASSIFICATION ON MINI-IMAGENET

We extend our investigation and assess SketchEmbeddings for the classification of natural images in the mini-ImageNet benchmark (Vinyals et al., 2016). The same CNN encoder model from the natural image sketching task is used to match the visual domain of the examples we hope to classify.

The mini-ImageNet (Vinyals et al., 2016) dataset consists of 100 classes each with 600 examples. The setup proposed by Ravi & Larochelle (2017) is used, where the classes are split 64-16-20 for training, validation and test. As noted earlier, any examples in the Sketchy dataset that are also present in the mini-ImageNet test were removed by filtering the synset (and children synset) IDs ensuring train and test classes are disjoint.

Classification results on mini-ImageNet are shown in Table 2. Natural image classification is a far more challenging problem. Learning to reconstruct pixels of an image actually worsens our results; the trained Conv-VAE is outperformed by the VAE with random weights. However, sketch reconstruction is still a valuable task; our models are competitive while our best model out-performs previous state-of-the-art unsupervised methods on few-shot settings. A full table is in Appendix J, seeding results are in Appendix F.

## 5.3 SKETCHING TO LEARN CLASS-IDENTIFIABLE INFORMATION

Existing sketch works have focused on generating better drawings or unifying sketches with other image domains. We present a new paradigm: using sketching as an auxiliary task to learn visual content. Only by training a drawing model that can sketch general image inputs are we able to transfer the learned understanding to new data distributions. By considering the stroke distribution of the Quickdraw dataset, we are able to interpret image inputs from the separate Omniglot dataset and tackle the few-shot classification task with surprising accuracy.

While the natural image sketching task is challenging and does not always produce high-fidelity results, it still learns useful visual information. By training on the Sketchy dataset, we learn how to draw other data distributions for which no sequential stroke data exists. Then, by knowing how to sketch this mini-ImageNet data we are able to produce distinguishable embeddings that enable competitive few-shot classification performance.

**Varying weighting of pixel-loss** For both settings we sweep the pixel loss coefficient $\alpha_{max}$ to ablate its impact on model performance on the Omniglot task (Table 3). There is a substantial improvement in few-shot classification when $\alpha_{max}$ is non-zero. $\alpha_{max}= 0.50$ achieves the best results, and the trend goes downward when $\alpha_{max}$ approaches to 1.0, i.e. the weighting for $\mathcal{L}_{stroke}$ goes to 0.0. This is

Figure 6: One-shot Omniglot generation compared to Rezende et al. (2016); Reed et al. (2017).

reasonable as the training of SketchEmbedNet is more stable under the guidance of ground truth strokes.

# 6 PROPERTIES OF SKETCHEMBEDDINGS

We hypothesize that reproducing a sketch drawing rather than a pixel-based approach requires the preservation of more structural information due to sequential RNN generation. By learning in this manner, SketchEmbeddings are aware of spatial properties and the composition of elements in image space. We examine this compositionality through several comparisons of SketchEmbeddings with those generated by a Conv-VAE.

**Component arrangements** We construct examples that contain the same set of objects but in different arrangements to test sensitivity to component arrangement and composition in image space. We then embed these examples with both generative models and project into 2D space using UMAP (McInnes et al., 2018) to visualize their organization. In the first 2 panels of Figure 5-A, we see that the SketchEmbeddings are easily separated in unsupervised clustering. The rightmost panel of Figure 5-A exhibits non-synthetic classes with duplicated shapes; snowmen with circles and televisions with squares. With these results, we demonstrate the greater component level awareness of SketchEmbeddings. The 4 rearranged shapes and the nested circle and squares have similar silhouettes that are difficult to differentiate to a conventional pixel loss. To SketchEmbeddings, the canvas offset and different drawing sequence of each shape make them substantially different in embedding space.

**Spatial relationships** Drawing also builds awareness of relevant underlying variables, such as spatial relationships between components of the image. We examine the degree to which the underlying variables of angle, distance or size are captured by the embedding, by constructing images that vary along each dimension respectively. The embeddings are again grouped by a 2D projection in Figure 5-B using the UMAP (McInnes et al., 2018) algorithm. When clustered, the 2D projection of SketchEmbeddings arranges the examples along an axis corresponding to the latent variable compared to the Conv-VAE embeddings which is visibly non-linear and arranges in clusters. This clear axis-alignment suggests a greater level of latent variable disentanglement in the SketchEmbeddings.

**Conceptual composition** Finally, we explore concept space composition using SketchEmbeddings (Figure 5-C) by embedding different Quickdraw examples then performing arithmetic with the latent vectors. By subtracting a circle embedding and adding a square embedding from a snowman composed of stacked circles, we produce stacked boxes. This property of vector arithmetic is reminiscent of language representations, as evidenced in analogies like King - Man + Woman = Queen (Ethayarajh et al., 2019). Our results indicate that this property is captured to a greater degree in SketchEmbedding than in the pixel-based VAE embeddings. Composing SketchEmbeddings produces decoded examples that appeal to our intuitive conceptual understanding while the VAE degrades to blurry, fragmented images. We provide more examples of the setup in Figure 5-C as well as additional classes in Appendix K.

# 7 ONE-SHOT GENERATION

To evaluate the sketches generated by our model, we make qualitative comparisons to other one-shot generative models and quantitatively assess our model through visual classification via a ResNet101 (He et al., 2016). In this section, all models use the ResNet12 (Oreshkin et al., 2018) backbone.

**Qualitative comparisons** We compare SketchEmbedNet one-shot generations of Omniglot characters with examples from other few-shot (Reed et al., 2017) and one-shot (Rezende et al., 2016)

Table 3: Effect of pixel loss coefficient $\alpha$ on Omniglot few-shot classification

| $\alpha_{max}$ | 20-way 1-shot Acc. |
|---|---|
| 0.00 | $87.17 \pm 0.36$ |
| 0.25 | $87.82 \pm 0.36$ |
| 0.50 | $\mathbf{91.39} \pm 0.31$ |
| 0.75 | $90.59 \pm 0.32$ |
| 0.95 | $89.77 \pm 0.32$ |

Table 4: ResNet-101 45-way classification score on 1-shot generated sketches of seen and unseen classes.

| Generation Method | Seen | Unseen |
|---|---|---|
| Original Data | 97.66 | 96.09 |
| Conv-VAE | $76.28 \pm 0.93$ | $75.07 \pm 0.84$ |
| SketchEmbedNet | $\mathbf{81.44} \pm 0.95$ | $\mathbf{77.94} \pm 1.07$ |

approaches (Figure 6). In the settings shown, none of the models have seen any examples from the character class, or the parent alphabet. Furthermore, the drawer has seen no written characters during training and is trained only on the Quickdraw dataset. Visually, our generated images better resemble the support examples and the variations by stretching and shifting strokes better preserves the semantics of each character. Generations in pixel space may disrupt strokes and alter the character to human perception. This is especially true for written characters as they are frequently defined by a specific set of strokes instead of blurry clusters of pixels.

**Quantitative evaluation of generation quality** Evaluating generative models is often challenging. Per-pixel metrics like in Reed et al. (2017); Rezende et al. (2016) may penalize generative variance that still preserves meaning. We propose an Inception Score (Salimans et al., 2016) inspired metric to quantify class-recognizability and generalization of generated examples. We train two separate ResNet classifiers (He et al., 2016), each on a different set of 45 Quickdraw classes. One set was part of the training set of SketchEmbedNet (referred to as "seen") and the other set was held out during training (referred to as "unseen"). We then have SketchEmbedNet generate one-shot sketches from each set and have the corresponding classifier predict a class. The accuracy of the classifier on generated examples is compared with its training accuracy in Table 4. For a ResNet classifier, SketchEmbedNet generations are more recognizable for both classes seen and unseen classes.

## 8 CONCLUSION

Learning to draw is not only an artistic pursuit but drives a distillation of real-world visual concepts. We present a generalized drawing model capable of producing accurate sketches and visual summaries of open-domain natural images. While sketch data may be challenging to source, we show that training to draw either sketch or natural images can generalize for downstream tasks, not only within each domain but also well beyond the training data. More generally research in this direction may lead to more lifelike image understanding inspired by how humans communicate visual concepts.

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

# A   RASTERIZATION

The key enabler of our novel pixel loss for sketch drawings is our differentiable rasterization function $f_{\text{raster}}$. Sequence based loss functions such as $\mathcal{L}_{\text{stroke}}$ are sensitive to the order of points while in reality, drawings are sequence invariant. Visually, a square is a square whether it is drawn clockwise or counterclockwise.

The purpose of a sketch representation is to lower the complexity of the data space and decode in a more visually intuitive manner. While it is a necessary departure point, the sequential generation of drawings is not key to our visual representation and we would like SketchEmbedNet to be agnostic to any specific sequence needed to draw the sketch that is representative of the image input.

To facilitate this, we develop our rasterization function $f_{\text{raster}}$ which renders an input sequence of strokes as a pixel image. However, during training, the RNN outputs a mixture of Gaussians at each timestep. To convert this to a stroke sequence, we sample from these Gaussians; this can be repeated to reduce the variance of the pixel loss. We then scale our predicted and ground truth sequences by the properties of the latter before rasterization.

**Stroke sampling**   At the end of sequence generation we have $N_s \times (6M+3)$ parameters, 6 Gaussian mixture parameters, 3 pen states, $N_s$ times, one for each stroke. To obtain the actual drawing we sample from the mixture of Gaussians:

$$\begin{bmatrix} \Delta x_t \\ \Delta y_t \end{bmatrix} = \begin{bmatrix} \mu_{x,t} \\ \mu_{y,t} \end{bmatrix} + \begin{bmatrix} \sigma_{x,t} & 0 \\ \rho_{xy,t}\sigma_{y,t} & \sigma_{y,t}\sqrt{1 - \rho_{xy,t}^2} \end{bmatrix} \boldsymbol{\epsilon} \,, \boldsymbol{\epsilon} \sim \mathcal{N}(\mathbf{0}, \mathbf{1}_2). \tag{3}$$

After sampling we compute the cumulative sum of every stroke over the timestep so that we obtain the absolute displacement from the initial position:

$$\begin{bmatrix} x_t \\ y_t \end{bmatrix} = \sum_{\tau=0}^{T} \begin{bmatrix} \Delta x_\tau \\ \Delta y_\tau \end{bmatrix}. \tag{4}$$

$$\boldsymbol{y}_{t,abs} = (x_t, y_t, s_1, s_2, s_3). \tag{5}$$

**Scaling**   Each sketch generated by our model begins at (0,0) and the variance of all strokes in the training set is normalized to $1$. On a fixed canvas the image is both very small and localized to the top left corner. We remedy this by computing a scale $\lambda$ and shift $x_{\text{shift}}, y_{\text{shift}}$ using labels $\boldsymbol{y}$ and apply them to both the prediction $\boldsymbol{y}'$ as well as the ground truth $\boldsymbol{y}$. These parameters are computed as:

$$\lambda = \min \left\{ \frac{W}{x_{\max} - x_{\min}}, \frac{H}{y_{\max} - y_{\min}} \right\}, \tag{6}$$

$$x_{\text{shift}} = \frac{x_{\max} + x_{\min}}{2}\lambda, \quad y_{\text{shift}} = \frac{y_{\max} + y_{\min}}{2}\lambda. \tag{7}$$

$x_{\max}, x_{\min}, y_{\max}, y_{\min}$ are the minimum and maximum values of $x_t, y_t$ from the supervised stroke labels and not the generated strokes. $W$ and $H$ are the width and height in pixels of our output canvas.

**Calculate pixel intensity**   Finally we are able to calculate the pixel $p_{ij}$ intensity of every pixel in our $H \times W$ canvas.

$$p_{ij} = \sigma \left[ 2 - 5 \times \min_{t=1\ldots N_s} \left( \text{dist}\big((i,j), (x_{t-1}, y_{t-1}), (x_t, y_t)\big) + (1 - \lfloor s_{1,t-1} \rfloor)10^6 \right) \right], \tag{8}$$

where the distance function is the distance between point $(i,j)$ from the line segment defined by the absolute points $(x_{t-1}, y_{t-1})$ and $(x_t, y_t)$. We also blow up any distances where $s_{1,t-1} < 0.5$ so as to not render any strokes where the pen is not touching the paper.

## B  Implementation details

We train our model for 300k iterations with a batch size of 256 for the Quickdraw dataset and 64 for Sketchy due to memory constraints. The initial learning rate is 1e-3 which decays by $0.85$ every 15k steps. We use the Adam (Kingma & Ba, 2015) optimizer and clip gradient values at 1.0. $\sigma = 2.0$ is used for the Gaussian blur in $\mathcal{L}_{\text{pixel}}$. For the curriculum learning schedule, the value of $\alpha$ is set to 0 initially and increases by 0.05 every 10k training steps with an empirically obtained cap at $\alpha_{\max} = 0.50$ for Quickdraw and $\alpha_{\max} = 0.75$ for Sketchy.

The ResNet12 (Oreshkin et al., 2018) encoder uses 4 ResNet blocks with 64, 128, 256, 512 filters respectively and ReLU activations. The Conv4 backbone has 4 blocks of convolution, batch norm (Ioffe & Szegedy, 2015), ReLU and max pool, identical to Vinyals et al. (2016). We select the latent space to be 256 dimensions, RNN output size to be 1024, and the hypernetwork embedding size to be 64. We use a mixture of $M = 30$ bivariate Gaussians for the mixture density output of the stroke offset distribution.

## C  Latent Space Interpolation

Like in many encoding-decoding models we evaluate the interpolation of our latent space. We select 4 embeddings at random and use bi-linear interpolation to produce new embeddings. Results are in Figures 7a and 7b.

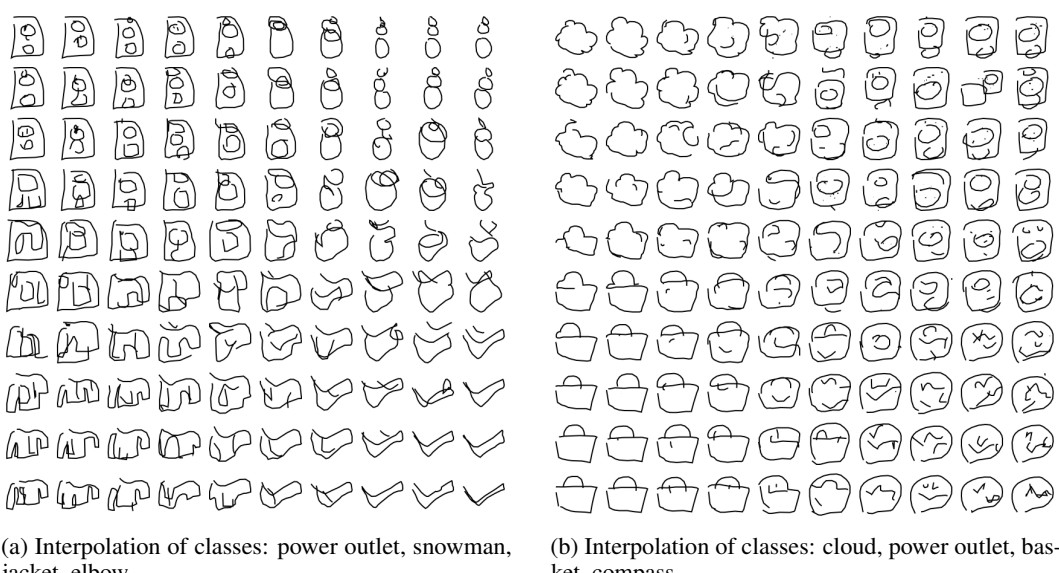

(a) Interpolation of classes: power outlet, snowman, jacket, elbow

(b) Interpolation of classes: cloud, power outlet, basket, compass

Figure 7: Latent space interpolations of randomly selected examples

We observe that compositionality is also present in these interpolations. In the top row of Figure 7a, the model first plots a third small circle when interpolating from the 2-circle power outlet and the 3-circle snowman. This small circle is treated as single component that grows as it transitions between classes until it's final size in the far right snowman drawing.

Some other RNN-based sketching models (Ha & Eck, 2018; Chen et al., 2017) experience other classes materializing in interpolations between two unrelated classes. Our model does not exhibit this same behaviour as our embedding space is learned from more classes and thus does not contain local groupings of classes.

# D    EFFECT OF $\alpha$ ON FEW-SHOT CLASSIFICATION

We performed additional experiments exploring the impact of our curriculum training schedule for $\alpha$. The encoding component of our drawing model was evaluated on the few-shot classification task for different values of $\alpha_{\max}$ every 25k iterations during training. A graph is shown in Figure 8 and the full table of all values of $\alpha_{\max}$ is in Table 5.

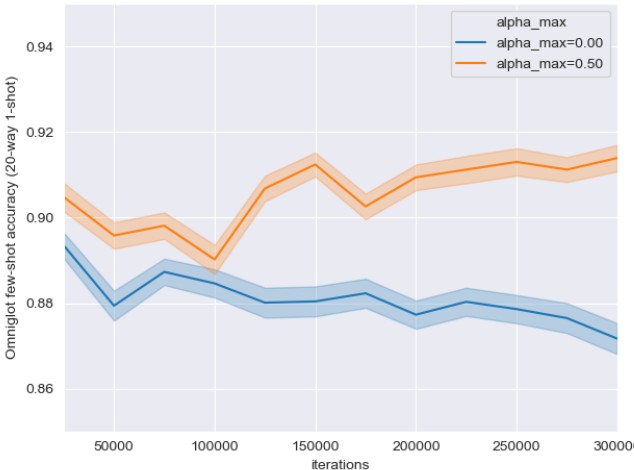

Figure 8: Few-shot classification accuracy of $\alpha_{\max}$ values 0.0 and 0.5 over training

Table 5: Few-shot classification accuracy of all $\alpha_{\max}$ values

| $\alpha_{\max}$ | 25k | 50k | 75k | 100k | 125k | 150k | 175k | 200k | 225k | 250k | 275k | 300k |
|---|---|---|---|---|---|---|---|---|---|---|---|---|
| 0.00 | 89.35 | 87.94 | 88.73 | 88.46 | 88.01 | 88.04 | 88.23 | 87.73 | 88.03 | 87.86 | 87.65 | 87.17 |
| 0.25 | 89.21 | 90.39 | 90.20 | 89.75 | 87.78 | 88.37 | 88.64 | 88.05 | 87.98 | 88.41 | 88.15 | 87.82 |
| 0.50 | 90.48 | 89.58 | 89.81 | 89.02 | 90.68 | 91.24 | 90.26 | 90.94 | 91.12 | 91.30 | 91.12 | 91.39 |
| 0.75 | 91.39 | 89.95 | 89.56 | 89.81 | 89.95 | 90.79 | 91.02 | 91.09 | 91.82 | 90.76 | 91.42 | 90.59 |
| 0.95 | 90.23 | 90.15 | 90.10 | 89.55 | 90.27 | 92.37 | 92.27 | 90.29 | 91.58 | 91.02 | 89.73 | 89.77 |

# E    INTRA-ALPHABET LAKE SPLIT

The creators of the Omniglot dataset and one-shot classification benchmark originally proposed an intra-alphabet classification task. This task is more challenging than the common Vinyals split as characters from the same alphabet may exhibit similar stylistics of sub-components that makes visual differentiation more difficult. This benchmark has been less explored by researchers; however, we still present the performance of our SketchEmbedding model against evaluations of other few-shot classification models on the benchmark. Results are shown in Table 6.

Table 6: Few-shot classification results on Omniglot (Lake split)

| Omniglot (Lake split) | | | (way, shot) | | | |
|---|---|---|---|---|---|---|
| Algorithm | Backbone | Train Data | (5,1) | (5,5) | (20,1) | (20,5) |
| Conv-VAE | Conv4 | Quickdraw | $73.12 \pm 0.58$ | $88.50 \pm 0.39$ | $53.45 \pm 0.51$ | $73.62 \pm 0.48$ |
| SketchEmbedding *(Ours)* | Conv4 | Quickdraw | $89.16 \pm 0.41$ | $97.12 \pm 0.18$ | $74.24 \pm 0.48$ | $89.87 \pm 0.25$ |
| SketchEmbedding *(Ours)* | ResNet12 | Quickdraw | $\mathbf{91.03 \pm 0.37}$ | $\mathbf{97.91 \pm 0.15}$ | $\mathbf{77.94 \pm 0.44}$ | $\mathbf{92.49 \pm 0.21}$ |
| BPL *(Supervised)* (Lake et al., 2015; 2019) | N/A | Omniglot | - | - | 96.70 | - |
| ProtoNet *(Supervised)* (Snell et al., 2017; Lake et al., 2019) | Conv4 | Omniglot | - | - | 86.30 | - |
| RCN *(Supervised)* (George et al., 2017; Lake et al., 2019) | N/A | Omniglot | - | - | 92.70 | - |
| VHE *(Supervised)* (Hewitt et al., 2018; Lake et al., 2019) | N/A | Omniglot | - | - | 81.30 | - |

Unsurprisingly, our model is outperformed by supervised models and does fall behind by a more substantial margin than in the Vinyals split. However, our SketchEmbedding approach still achieves respectable classification accuracy overall and greatly outperforms a Conv-VAE baseline.

## F EFFECT OF RANDOM SEEDING ON FEW-SHOT CLASSIFICATION

The training objective for SketchEmbedNet is to reproduce sketch drawings of the input. This task is unrelated to few-shot classification may perform variably given different initialization. We quantify this variance by training our model with 15 unique random seeds and evaluating the performance of the latent space on the few-shot classification tasks.

We disregard the per (evaluation) episode variance of our model in each test stage and only present the mean accuracy. We then compute a new confidence interval over random seeds. Results are presented in Tables 7, 8, 9.

Table 7: Random Seeding on Few-Shot Classification results on Omniglot (Conv4)

| | (way, shot) | | | |
|---|---|---|---|---|
| Seed | (5,1) | (5,5) | (20,1) | (20,5) |
| 1 | 96.45 | 99.41 | 90.84 | 98.08 |
| 2 | 96.54 | 99.48 | 90.82 | 98.10 |
| 3 | 96.23 | 99.40 | 90.05 | 97.94 |
| 4 | 96.15 | 99.46 | 90.50 | 97.99 |
| 5 | 96.21 | 99.40 | 90.54 | 98.10 |
| 6 | 96.08 | 99.43 | 90.20 | 97.93 |
| 7 | 96.19 | 99.39 | 90.70 | 98.05 |
| 8 | 96.68 | 99.44 | 91.11 | 98.18 |
| 9 | 96.49 | 99.42 | 90.64 | 98.06 |
| 10 | 96.37 | 99.47 | 90.50 | 97.99 |
| 11 | 96.52 | 99.40 | 91.13 | 98.18 |
| 12 | **96.96** | **99.50** | **91.67** | **98.30** |
| 13 | 96.31 | 99.38 | 90.57 | 98.04 |
| 14 | 96.12 | 99.45 | 90.54 | 98.03 |
| 15 | 96.30 | 99.48 | 90.62 | 98.05 |
| Average | $96.37 \pm 0.12$ | $99.43 \pm 0.02$ | $90.69 \pm 0.20$ | $98.07 \pm 0.05$ |

Table 8: Random Seeding on Few-Shot Classification results on Omniglot (ResNet12)

| | (way, shot) | | | |
|---|---|---|---|---|
| Seed | (5,1) | (5,5) | (20,1) | (20,5) |
| 1 | **96.61** | **99.58** | **91.25** | **98.58** |
| 2 | 96.37 | 99.52 | 90.44 | 98.40 |
| 3 | 96.04 | 99.58 | 89.86 | 98.27 |
| 4 | 96.44 | 99.50 | 90.76 | 98.40 |
| 5 | 95.95 | 99.52 | 89.88 | 98.29 |
| 6 | 95.63 | 99.45 | 89.28 | 98.17 |
| 7 | 96.24 | 99.52 | 89.90 | 98.34 |
| 8 | 95.41 | 99.45 | 88.75 | 98.05 |
| 9 | 96.04 | 99.49 | 89.70 | 98.24 |
| 10 | 95.40 | 99.41 | 88.91 | 98.05 |
| 11 | 95.82 | 99.51 | 89.67 | 98.24 |
| 12 | 96.25 | 99.51 | 90.21 | 98.28 |
| 13 | 95.84 | 99.53 | 89.71 | 98.18 |
| 14 | 96.04 | 99.56 | 89.89 | 98.31 |
| 15 | 96.04 | 99.57 | 89.97 | 98.32 |
| Average | $96.00 \pm 0.31$ | $99.51 \pm 0.04$ | $89.89 \pm 0.56$ | $98.27 \pm 0.12$ |

Table 9: Random Seeding on Few-Shot Classification results on mini-ImageNet

| | (way, shot) | | | |
|---|---|---|---|---|
| **Seed** | **(5,1)** | **(5,5)** | **(5,20)** | **(5,50)** |
| 1 | 37.15 | 52.99 | 63.92 | 68.72 |
| 2 | 39.38 | 55.20 | 65.60 | 69.79 |
| 3 | 39.40 | 55.47 | 65.94 | 70.41 |
| 4 | **40.39** | **57.15** | **67.60** | **71.99** |
| 5 | 38.40 | 54.08 | 65.36 | 70.08 |
| 6 | 37.94 | 53.98 | 65.24 | 69.65 |
| 7 | 38.88 | 55.71 | 66.59 | 71.35 |
| 8 | 37.89 | 52.65 | 63.42 | 68.14 |
| 9 | 38.25 | 53.86 | 65.02 | 69.82 |
| 10 | 39.11 | 55.29 | 65.99 | 69.98 |
| 11 | 37.39 | 52.88 | 63.66 | 68.33 |
| 12 | 38.24 | 53.91 | 65.19 | 69.82 |
| 13 | 38.62 | 53.84 | 63.83 | 68.69 |
| 14 | 37.73 | 53.61 | 64.22 | 68.41 |
| 15 | 39.50 | 55.23 | 65.51 | 70.25 |
| Average | $38.55 \pm 0.45$ | $54.39 \pm 0.63$ | $65.14 \pm 0.59$ | $69.69 \pm 0.56$ |

# G   DATA PROCESSING

## G.1   QUICKDRAW

We apply the same data processing methods as in Ha & Eck (2018) with no additional changes to produce our stroke labels $y$. When rasterizing for our input $x$, we scale, center the strokes then pad the image with 10% of the resolution in that dimension rounded to the nearest integer.

The following list of classes were used for training: The Eiffel Tower, The Mona Lisa, aircraft carrier, alarm clock, ambulance, angel, animal migration, ant, apple, arm, asparagus, banana, barn, baseball, baseball bat, bathtub, beach, bear, bed, bee, belt, bench, bicycle, binoculars, bird, blueberry, book, boomerang, bottlecap, bread, bridge, broccoli, broom, bucket, bulldozer, bus, bush, butterfly, cactus, cake, calculator, calendar, camel, camera, camouflage, campfire, candle, cannon, car, carrot, castle, cat, ceiling fan, cell phone, cello, chair, chandelier, church, circle, clarinet, clock, coffee cup, computer, cookie, couch, cow, crayon, crocodile, crown, cruise ship, diamond, dishwasher, diving board, dog, dolphin, donut, door, dragon, dresser, drill, drums, duck, dumbbell, ear, eye, eyeglasses, face, fan, feather, fence, finger, fire hydrant, fireplace, firetruck, fish, flamingo, flashlight, flip flops, flower, foot, fork, frog, frying pan, garden, garden hose, giraffe, goatee, grapes, grass, guitar, hamburger, hand, harp, hat, headphones, hedgehog, helicopter, helmet, hockey puck, hockey stick, horse, hospital, hot air balloon, hot dog, hourglass, house, house plant, ice cream, key, keyboard, knee, knife, ladder, lantern, leaf, leg, light bulb, lighter, lighthouse, lightning, line, lipstick, lobster, mailbox, map, marker, matches, megaphone, mermaid, microphone, microwave, monkey, mosquito, motorbike, mountain, mouse, moustache, mouth, mushroom, nail, necklace, nose, octopus, onion, oven, owl, paint can, paintbrush, palm tree, parachute, passport, peanut, pear, pencil, penguin, piano, pickup truck, pig, pineapple, pliers, police car, pool, popsicle, postcard, purse, rabbit, raccoon, radio, rain, rainbow, rake, remote control, rhinoceros, river, rollerskates, sailboat, sandwich, saxophone, scissors, see saw, shark, sheep, shoe, shorts, shovel, sink, skull, sleeping bag, smiley face, snail, snake, snowflake, soccer ball, speedboat, square, star, steak, stereo, stitches, stop sign, strawberry, streetlight, string bean, submarine, sun, swing set, syringe, t-shirt, table, teapot, teddy-bear, tennis racquet, tent, tiger, toe, tooth, toothpaste, tractor, traffic light, train, triangle, trombone, truck, trumpet, umbrella, underwear, van, vase, watermelon, wheel, windmill, wine bottle, wine glass, wristwatch, zigzag, blackberry, power outlet, peas, hot tub, toothbrush, skateboard, cloud, elbow, bat, pond, compass, elephant, hurricane, jail, school bus, skyscraper, tornado, picture frame, lollipop, spoon, saw, cup, roller coaster, pants, jacket, rifle, yoga, toilet, waterslide, axe, snowman, bracelet, basket, anvil, octagon, washing machine, tree, television, bowtie, sweater, backpack, zebra, suitcase, stairs, The Great Wall of China

## G.2   OMNIGLOT

We derive our Omniglot tasks from the stroke dataset originally provided by Lake et al. (2015) rather than the image analogues. We translate the Omniglot stroke-by-stroke format to the same one used in Quickdraw. Then we apply the Ramer-Douglas-Peucker (Douglas & Peucker, 1973) algorithm with an epsilon value of 2 and normalize variance to 1 to produce $y$. We also rasterize our images in the same manner as above for our input $x$.

## G.3   SKETCHY

Sketchy data is provided as an SVG image composed of line paths that are either straight lines or Bezier curves. To generate stroke data we sample sequences of points from Bezier curves at a high resolution that we then simplify with RDP, $\epsilon = 5$. We also eliminate continuous strokes with a short

path length or small displacement to reduce our stroke length and remove small and noisy strokes. Path length and displacement are considered with respect to the scale of the entire sketch.

Once again we normalize stroke variance and rasterize for our input image in the same manners as above.

The following classes were use for training after removing overlapping classes with mini-ImageNet:

hot-air_balloon, violin, tiger, eyeglasses, mouse, jack-o-lantern, lobster, teddy_bear, teapot, helicopter, duck, wading_bird, rabbit, penguin, sheep, windmill, piano, jellyfish, table, fan, beetle, cabin, scorpion, scissors, banana, tank, umbrella, crocodilian, volcano, knife, cup, saxophone, pistol, swan, chicken, sword, seal, alarm_clock, rocket, bicycle, owl, squirrel, hermit_crab, horse, spoon, cow, hotdog, camel, turtle, pizza, spider, songbird, rifle, chair, starfish, tree, airplane, bread, bench, harp, seagull, blimp, apple, geyser, trumpet, frog, lizard, axe, sea_turtle, pretzel, snail, butterfly, bear, ray, wine_bottle, , elephant, raccoon, rhinoceros, door, hat, deer, snake, ape, flower, car_(sedan), kangaroo, dolphin, hamburger, castle, pineapple, saw, zebra, candle, cannon, racket, church, fish, mushroom, strawberry, window, sailboat, hourglass, cat, shoe, hedgehog, couch, giraffe, hammer, motorcycle, shark

## H AUTOREGRESSIVE DRAWING MODEL COMPARISONS

We summarize the key components of SketchEmbedNet in comparison to other autoregressive drawing models in Table 10.

Table 10: Model comparisons between generative autoregressive models that produce pixel or vector sketch drawings.

| Autoregressive sketching models | | | | | |
|---|---|---|---|---|---|
| Model | Dataset | # classes | Encoder | Decoder | Loss function |
| Handwriting Sequence Graves (2013) | IAM-OnDB Liwicki & Bunke (2005) | 1 | RNN | Mixture Density RNN | $\mathcal{L}_{\text{stroke}}$ |
| DRAW Gregor et al. (2015) | SVHN Netzer et al. (2011), MNIST LeCun et al. (1998) | 10 | RNN | RNN | $\mathcal{L}_{\text{pixel}} + \mathcal{L}_{\text{KL}}$ |
| Sketch-RNN Ha & Eck (2018) | Quickdraw Jongejan et al. (2016) | 1 | Bi-directional RNN | Mixture Density RNN | $\mathcal{L}_{\text{pen}} + \mathcal{L}_{\text{stroke}} + \mathcal{L}_{\text{KL}}$ |
| Sketch-pix2seq Chen et al. (2017) | Quickdraw Jongejan et al. (2016) | 3, 6 | simple CNN | Mixture Density RNN | $\mathcal{L}_{\text{pen}} + \mathcal{L}_{\text{stroke}}$ |
| AI-Sketcher Cao et al. (2019) | Quickdraw Jongejan et al. (2016), FaceX Cao et al. (2019) | 5, 10, 15, 20 | Bi-directional RNN + CNN Autoencoder | Mixture Density RNN | $\mathcal{L}_{\text{pen}} + \mathcal{L}_{\text{stroke}} + \mathcal{L}_{\text{KL}}$ |
| deep_p2s Song et al. (2018) | Quickdraw Jongejan et al. (2016), ShoesV2 Yu et al. (2016), ChairV2 | 1 | Bi-directional RNN, CNN | CNN, Mixture Density RNN | $\mathcal{L}_{\text{pen}} + \mathcal{L}_{\text{stroke}} + \mathcal{L}_{l2}$ $+ \mathcal{L}_{\text{KL}} + \mathcal{L}_{\text{shortcut}}$ |
| SketchEmbedding *(ours)* | Quickdraw Jongejan et al. (2016) | 300 | ResNet12 Oreshkin et al. (2018) | Mixture Density RNN | $\mathcal{L}_{\text{pen}} + \mathcal{L}_{\text{stroke}} + \mathcal{L}_{\text{pixel}}$ |

## I FEW-SHOT CLASSIFICATION ON OMNIGLOT – FULL RESULTS

The full results table for few-shot classification on the Omniglot (Lake et al., 2015) dataset, including the ResNet12 (Oreshkin et al., 2018) model.

Table 11: Few-shot classification results on Omniglot

| Omniglot | | | (way, shot) | | | |
| --- | --- | --- | --- | --- | --- | --- |
| Algorithm | Backbone | Train Data | (5,1) | (5,5) | (20,1) | (20,5) |
| Training from Scratch (Hsu et al., 2019) | N/A | Omniglot | $52.50 \pm 0.84$ | $74.78 \pm 0.69$ | $24.91 \pm 0.33$ | $47.62 \pm 0.44$ |
| Random CNN | Conv4 | N/A | $67.96 \pm 0.44$ | $83.85 \pm 0.31$ | $44.39 \pm 0.23$ | $60.87 \pm 0.22$ |
| Conv-VAE | Conv4 | Omniglot | $77.83 \pm 0.41$ | $92.91 \pm 0.19$ | $62.59 \pm 0.24$ | $84.01 \pm 0.15$ |
| Conv-VAE | Conv4 | Quickdraw | $81.49 \pm 0.39$ | $94.09 \pm 0.17$ | $66.24 \pm 0.23$ | $86.02 \pm 0.14$ |
| Conv-AE | Conv4 | Quickdraw | $81.54 \pm 0.40$ | $93.57 \pm 0.19$ | $67.24 \pm 0.24$ | $84.15 \pm 0.16$ |
| $\beta$-VAE ($\beta = 250$) (Higgins et al., 2017) | Conv4 | Quickdraw | $79.11 \pm 0.40$ | $93.23 \pm 0.19$ | $63.67 \pm 0.24$ | $84.92 \pm 0.15$ |
| k-NN (Hsu et al., 2019) | N/A | Omniglot | $57.46 \pm 1.35$ | $81.16 \pm 0.57$ | $39.73 \pm 0.38$ | $66.38 \pm 0.36$ |
| Linear Classifier (Hsu et al., 2019) | N/A | Omniglot | $61.08 \pm 1.32$ | $81.82 \pm 0.58$ | $43.20 \pm 0.69$ | $66.33 \pm 0.36$ |
| MLP + Dropout (Hsu et al., 2019) | N/A | Omniglot | $51.95 \pm 0.82$ | $77.20 \pm 0.65$ | $30.65 \pm 0.39$ | $58.62 \pm 0.41$ |
| Cluster Matching (Hsu et al., 2019) | N/A | Omniglot | $54.94 \pm 0.85$ | $71.09 \pm 0.77$ | $32.19 \pm 0.40$ | $45.93 \pm 0.40$ |
| CACTUs-MAML (Hsu et al., 2019) | Conv4 | Omniglot | $68.84 \pm 0.80$ | $87.78 \pm 0.50$ | $48.09 \pm 0.41$ | $73.36 \pm 0.34$ |
| CACTUs-ProtoNet (Hsu et al., 2019) | Conv4 | Omniglot | $68.12 \pm 0.84$ | $83.58 \pm 0.61$ | $47.75 \pm 0.43$ | $66.27 \pm 0.37$ |
| AAL-ProtoNet (Antoniou & Storkey, 2019) | Conv4 | Omniglot | $84.66 \pm 0.70$ | $88.41 \pm 0.27$ | $68.79 \pm 1.03$ | $74.05 \pm 0.46$ |
| AAL-MAML (Antoniou & Storkey, 2019) | Conv4 | Omniglot | $88.40 \pm 0.75$ | $98.00 \pm 0.32$ | $70.20 \pm 0.86$ | $88.30 \pm 1.22$ |
| UMTRA (Khodadadeh et al., 2019) | Conv4 | Omniglot | $83.80$ | $95.43$ | $74.25$ | $92.12$ |
| SketchEmbedding (Ours) | Conv4 | Omniglot | $94.88 \pm 0.22$ | $99.01 \pm 0.08$ | $86.18 \pm 0.18$ | $96.69 \pm 0.07$ |
| SketchEmbedding-avg (Ours) | Conv4 | Quickdraw | $96.37$ | $99.43$ | $90.69$ | $98.07$ |
| SketchEmbedding-best (Ours) | Conv4 | Quickdraw | $\mathbf{96.96} \pm 0.17$ | $\mathbf{99.50} \pm 0.06$ | $\mathbf{91.67} \pm 0.14$ | $98.30 \pm 0.05$ |
| SketchEmbedding-avg (Ours) | ResNet12 | Quickdraw | $96.00$ | $99.51$ | $89.88$ | $98.27$ |
| SketchEmbedding-best (Ours) | ResNet12 | Quickdraw | $96.61 \pm 0.19$ | $\mathbf{99.58} \pm 0.06$ | $91.25 \pm 0.15$ | $\mathbf{98.58} \pm 0.05$ |
| SketchEmbedding(KL)-avg (Ours) | Conv4 | Quickdraw | $96.06$ | $99.40$ | $89.83$ | $97.92$ |
| SketchEmbedding(KL)-best (Ours) | Conv4 | Quickdraw | $96.60 \pm 0.18$ | $99.46 \pm 0.06$ | $90.84 \pm 0.15$ | $98.09 \pm 0.06$ |
| SketchEmbedding (w/ Labels) (Ours) | Conv4 | Quickdraw | $88.52 \pm 0.34$ | $96.73 \pm 0.13$ | $71.35 \pm 0.24$ | $88.16 \pm 0.14$ |
| MAML (Supervised) (Finn et al., 2017) | Conv4 | Omniglot | $94.46 \pm 0.35$ | $98.83 \pm 0.12$ | $84.60 \pm 0.32$ | $96.29 \pm 0.13$ |
| ProtoNet (Supervised) (Snell et al., 2017) | Conv4 | Omniglot | $98.35 \pm 0.22$ | $99.58 \pm 0.09$ | $95.31 \pm 0.18$ | $98.81 \pm 0.07$ |

* Stroke data used for training

# J  FEW-SHOT CLASSIFICATION ON MINI-IMAGENET – FULL RESULTS

The full results table for few-shot classification on the mini-ImageNet dataset, including the ResNet12 (Oreshkin et al., 2018) model and Conv4 models.

Table 12: Few-shot classification results on mini-ImageNet

| mini-ImageNet | | | (way, shot) | | | |
| --- | --- | --- | --- | --- | --- | --- |
| Algorithm | Backbone | Train Data | (5,1) | (5,5) | (5,20) | (5,50) |
| Training from Scratch (Hsu et al., 2019) | N/A | mini-ImageNet | $27.59 \pm 0.59$ | $38.48 \pm 0.66$ | $51.53 \pm 0.72$ | $59.63 \pm 0.74$ |
| UMTRA (Khodadadeh et al., 2019) | Conv4 | mini-ImageNet | $39.93$ | $50.73$ | $61.11$ | $67.15$ |
| CACTUs-MAML (Hsu et al., 2019) | Conv4 | mini-ImageNet | $39.90 \pm 0.74$ | $53.97 \pm 0.70$ | $63.84 \pm 0.70$ | $69.64 \pm 0.63$ |
| CACTUs-ProtoNet (Hsu et al., 2019) | Conv4 | mini-ImageNet | $39.18 \pm 0.71$ | $53.36 \pm 0.70$ | $61.54 \pm 0.68$ | $63.55 \pm 0.64$ |
| AAL-ProtoNet (Antoniou & Storkey, 2019) | Conv4 | mini-ImageNet | $37.67 \pm 0.39$ | $40.29 \pm 0.68$ | - | - |
| AAL-MAML (Antoniou & Storkey, 2019) | Conv4 | mini-ImageNet | $34.57 \pm 0.74$ | $49.18 \pm 0.47$ | - | - |
| Random CNN | Conv4 | N/A | $26.85 \pm 0.31$ | $33.37 \pm 0.32$ | $38.51 \pm 0.28$ | $41.41 \pm 0.28$ |
| Conv-VAE | Conv4 | mini-ImageNet | $23.30 \pm 0.21$ | $26.22 \pm 0.20$ | $29.93 \pm 0.21$ | $32.57 \pm 0.20$ |
| Conv-VAE | Conv4 | Sketchy | $23.27 \pm 0.18$ | $26.28 \pm 0.19$ | $30.41 \pm 0.19$ | $33.97 \pm 0.19$ |
| Random CNN | ResNet12 | N/A | $28.59 \pm 0.34$ | $35.91 \pm 0.34$ | $41.31 \pm 0.33$ | $44.07 \pm 0.31$ |
| Conv-VAE | ResNet12 | mini-ImageNet | $23.82 \pm 0.23$ | $28.16 \pm 0.25$ | $33.64 \pm 0.27$ | $37.81 \pm 0.27$ |
| Conv-VAE | ResNet12 | Sketchy | $24.61 \pm 0.23$ | $28.85 \pm 0.23$ | $35.72 \pm 0.27$ | $40.44 \pm 0.28$ |
| SketchEmbedding-avg (ours) | Conv4 | Sketchy* | $37.01$ | $51.49$ | $61.41$ | $65.75$ |
| SketchEmbedding-best (ours) | Conv4 | Sketchy* | $38.61 \pm 0.42$ | $53.82 \pm 0.41$ | $63.34 \pm 0.35$ | $67.22 \pm 0.32$ |
| SketchEmbedding-avg (ours) | ResNet12 | Sketchy* | $38.55$ | $54.39$ | $65.14$ | $69.70$ |
| SketchEmbedding-best (ours) | ResNet12 | Sketchy* | $\mathbf{40.39} \pm 0.44$ | $\mathbf{57.15} \pm 0.38$ | $\mathbf{67.60} \pm 0.33$ | $\mathbf{71.99} \pm 0.3$ |
| MAML (supervised) (Finn et al., 2017) | Conv4 | mini-ImageNet | $46.81 \pm 0.77$ | $62.13 \pm 0.72$ | $71.03 \pm 0.69$ | $75.54 \pm 0.62$ |
| ProtoNet (supervised) (Snell et al., 2017) | Conv4 | mini-ImageNet | $46.56 \pm 0.76$ | $62.29 \pm 0.71$ | $70.05 \pm 0.65$ | $72.04 \pm 0.60$ |

* Stroke data used for training

## K  ADDITIONAL CONCEPTUAL COMPOSITIONALITY

Figure 9: Uncherrypicked conceptual compositionality examples

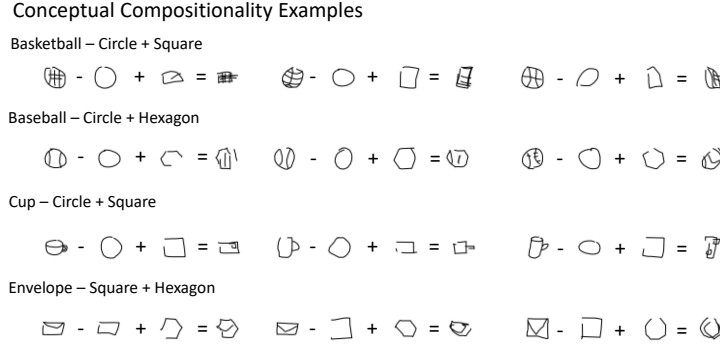

Figure 10: Additional conceptual compositionality examples

## L  EMBEDDING PROPERTIES OF OTHER BASELINE MODELS

Here we substantiate the uniqueness of the properties observed in SketchEmbeddings by applying the same experiments to a $\beta$-VAE (Higgins et al., 2017) as well a vanilla autoencoder trained on the same dataset. We also include results of a SketchEmbedNet trained with a KL objective.

## L.1 $\beta$-VAE

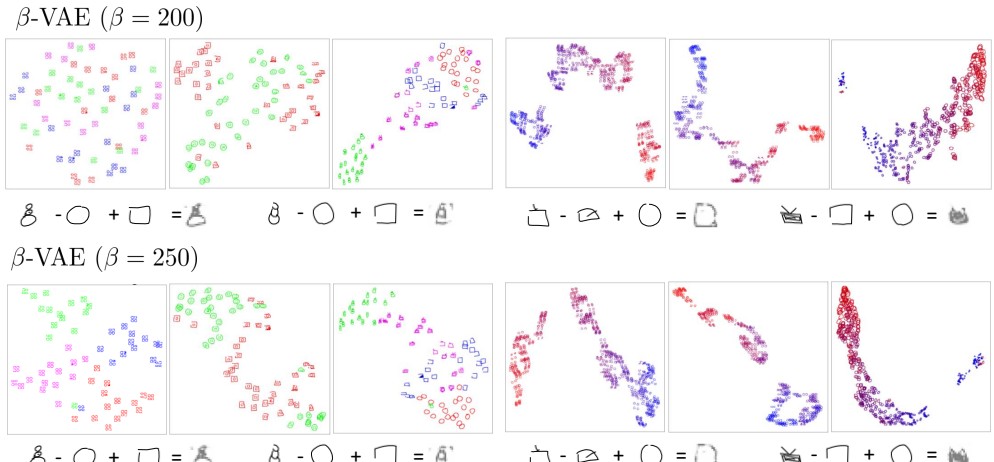

Figure 11: Section 6 results for $\beta$-VAE

The $\beta$-VAE (Higgins et al., 2017) exhibits similar unsupervised clustering in comparison to the Conv-VAE and is generally incapable of distinguishing input images that have different shape compositions but the same overall silhouette (first two examples from the left). Differently it is better at distinguishing non-synthetic examples that contain multiple squares or circles (3rd figure). However, it utterly fails the latent variable regression task and does not exhibit any significant conceptual composition in latent space.

## L.2 AUTOENCODER AND SKETCHEMBEDNET-KL

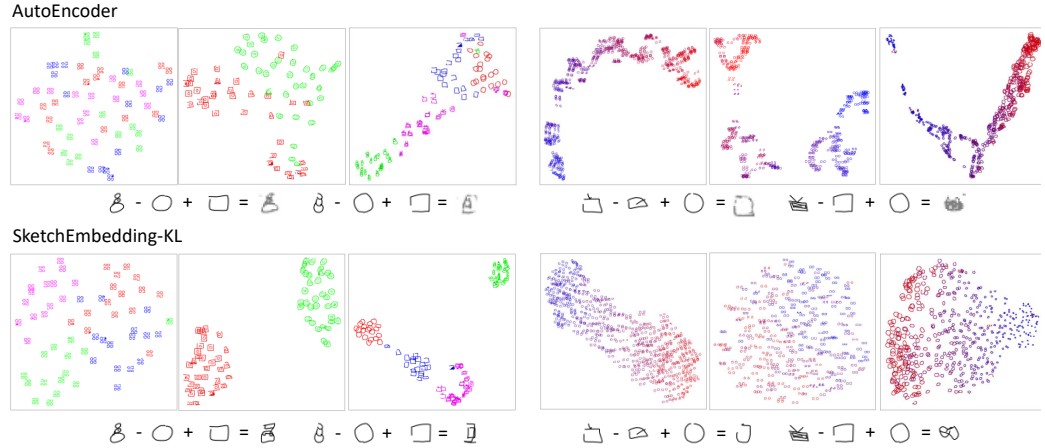

Figure 12: Section 6 results for Autoencoder and SketchEmbedding-KL

We show that the performance of SketchEmbedding embeddings in our experiments in Section 6 which focuses on organization in latent space is not correlated with the KL term. We present both a vanilla autoencoder without the KL objective and a SketchEmbedNet trained with a KL objective. We observe a drop in overall generation quality in the Conceptual Composition decoding as is expected with an additional constraint but maintained performance in the other tasks. Meanwhile, the autoencoder does not demonstrate any marked improvements over the Conv-VAE in the main paper or any other baseline comparison.

# M    ADDITIONAL COMPOSITIONALITY MODES

We provide additional clustering methods t-SNE (Maaten & Hinton, 2008) and PCA as well as 2 new experiments that explore the compositionality of our latent SketchEmbedding.

**Additional clustering methods**    We include additional t-SNE and PCA results of the experiments in the main paper. These are presented in Figures 13, 14, 15 16, 17. t-SNE and UMAP are stochastic and do not always produce the same visualization while PCA is deterministic and prioritizes the most important dimensions.

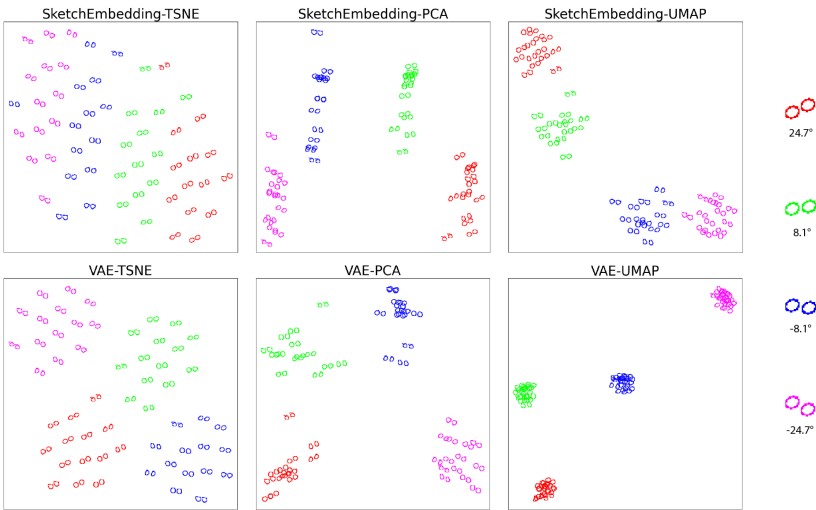

Figure 13: 2D Embedding visualization of different spatial orientations of circles and squares

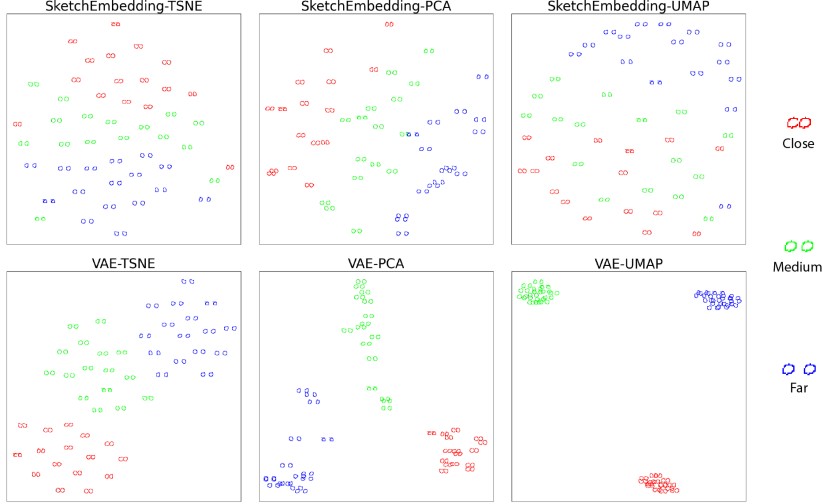

Figure 14: 2D Embedding visualization of different linear distances between shapes

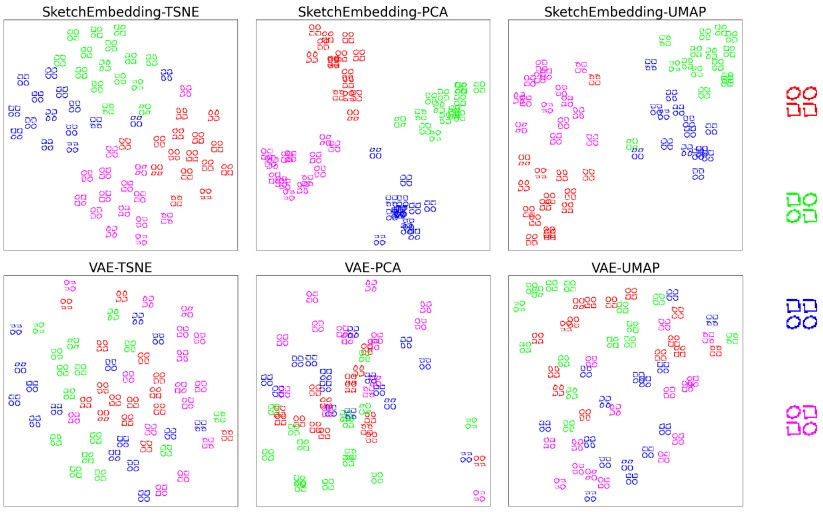

Figure 15: Latent space visualization squares and circles arranged differently in a 2x2 array

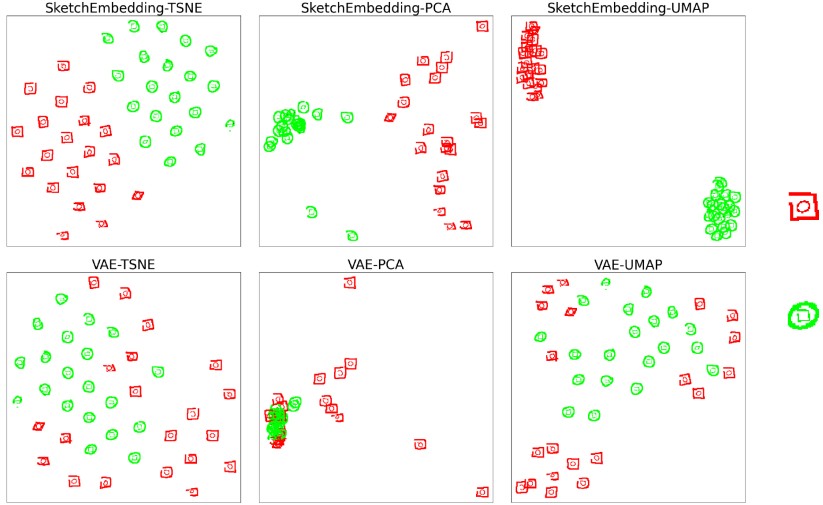

Figure 16: Latent space visualization of composing circles and squares within one another or outside

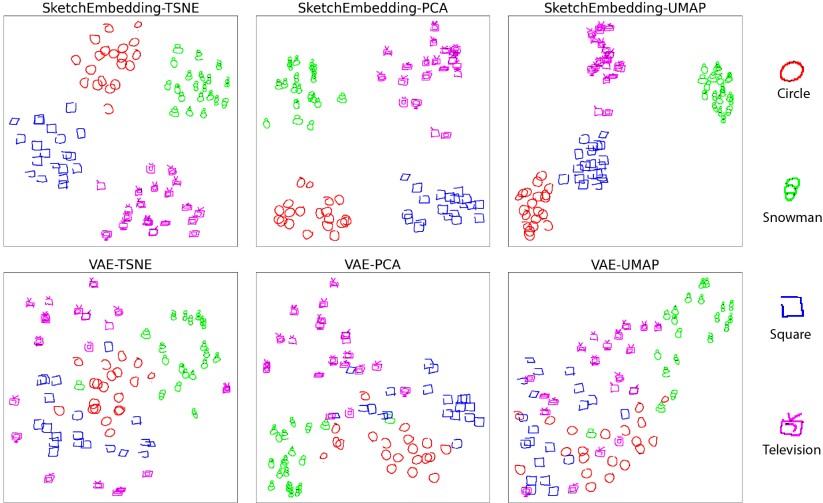

Figure 17: Latent space visualization of composing multiple circles and squares in real sketch drawings

**Additional Experiments** Here we provide different investigations into the compositionality of our learned embedding space that were not present in our main paper. These results presented in Figure 18 and 19.

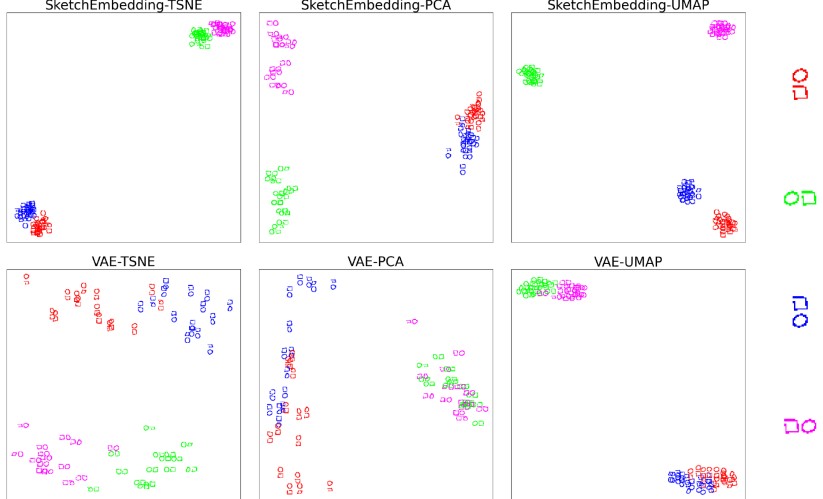

Figure 18: 2D Embedding visualization of different spatial orientations of circles and squares

In Figure 18 we place a square in the center of the example and place a circle above, below or to the sides of it. Once again we find that our SketchEmbedding embedding clusters better than the VAE approach.

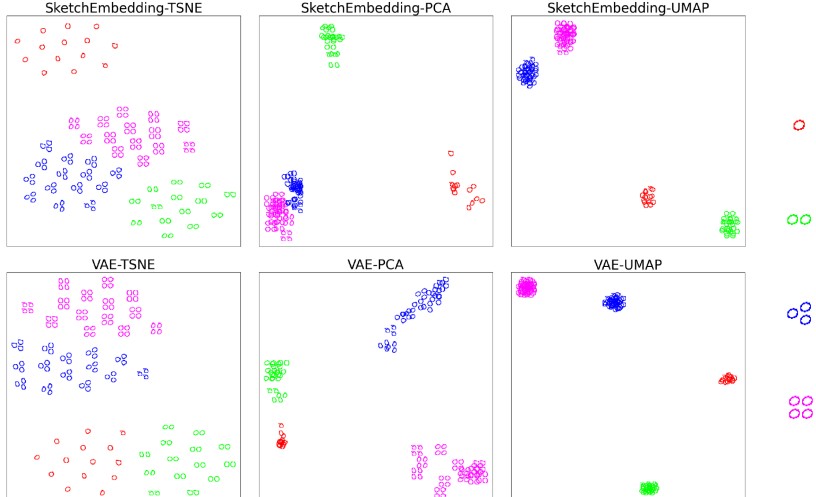

Figure 19: Latent space visualization of composing multiple circles and squares in real sketch drawings

New examples are generated where each class has a different numbers of circles. Both the VAE approach and our SketchEmbedding cluster well and neither appear to learn the count manifold.

## N  HYPERNETWORK ACTIVATIONS

To further explore how our network understands drawings, we examine the relationships between the activations of the hypernetwork of our HyperLSTM (Ha et al., 2017).

The hypernetwork determines the weights of the LSTM that generates the RNN at each decoding timestep. These activations are 512-dimensional vectors. We collect the activations from many

examples, cluster them in 512-dimensional space and visualize the strokes belonging to each cluster for each example. A full decoding is also rendered where each cluster within an example is assigned a color.

**Single class: snowman** First we explore this clustering using only the snowman class from Quick-draw (Jongejan et al., 2016). We expect substantial reuse of a "circle" both within and over many examples. Clustering of the strokes is done with the DBSCAN (Ester et al., 1996) and parameter $\epsilon = 3.9$. Results are in Figure 20. Each row is a separate input; the far left column is the color-coded, composed image, the second is the noise cluster and every subsequent column is a unique cluster.

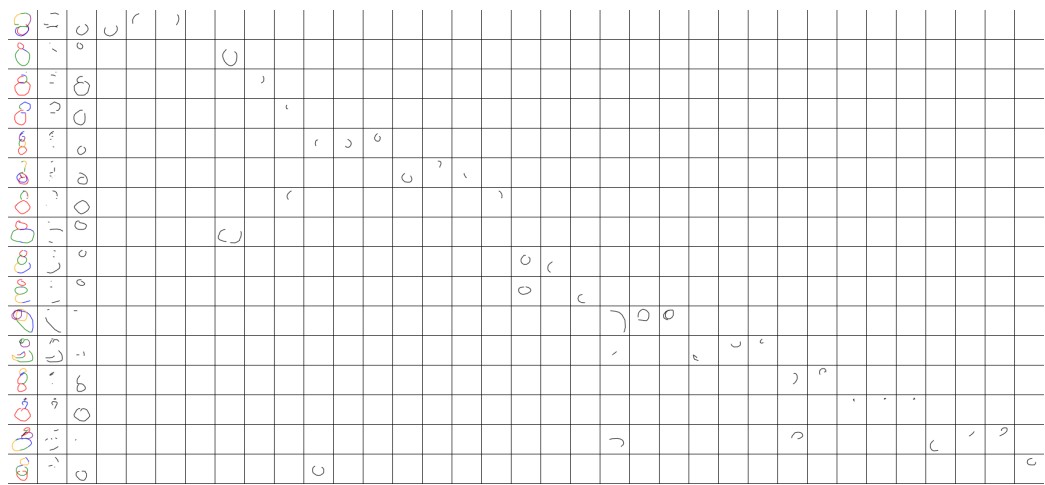

Figure 20: Snowman class stroke clustering

While cluster re-use is limited, cluster 0 often contains a large, fully enclosed circle. Many other clusters may contain circles or partial strokes with some reuse. Larger, fully composed and coloured sketches are presented in Figure 21

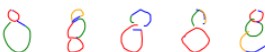

Figure 21: Fully composed images with coloured cluster assignments

**Many classes: round objects** We repeat the above experiment with a mixture of classes that generally can be expected to contain circles. These classes were circles, snowmen, clocks and cups. The two former classes are frequently composed only of circles while the latter are expected to consistently contain other distinct shapes. Results are presented in Figure 22 and select examples in Figure 23.

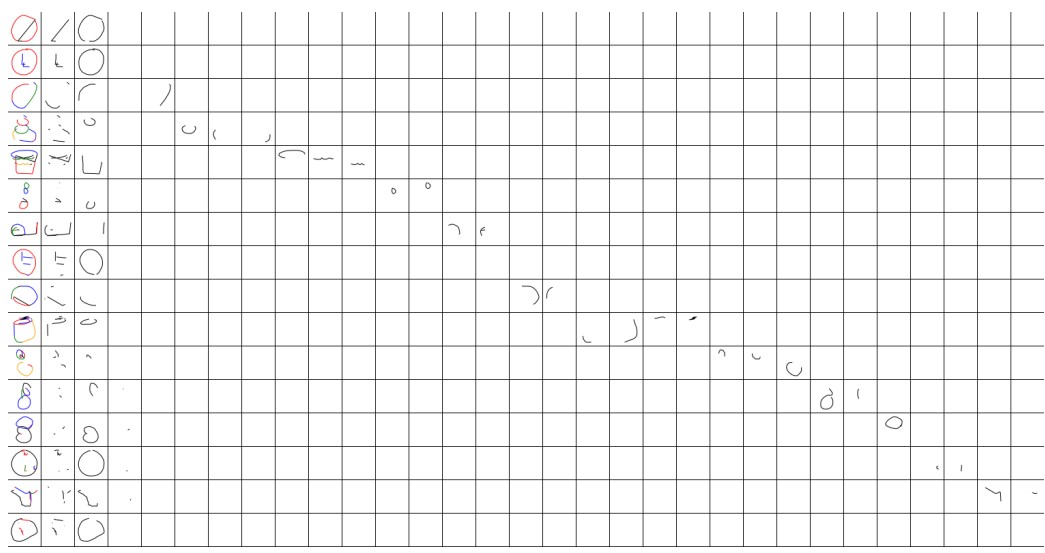

Figure 22: Snowman class stroke clustering

We still observe that the model continues to isolate circles in the first column and note it continues to do so for the cup and clock classes which are not exclusively circular.

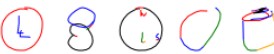

Figure 23: Fully composed images with coloured cluster assignments

**Many random classes:** Finally, we repeat the above clustering with the 45 randomly selected holdout classes from the Quickdraw training process of SketchEmbedding. Results are once again presented in Figure 24 and select examples in Figure 25.

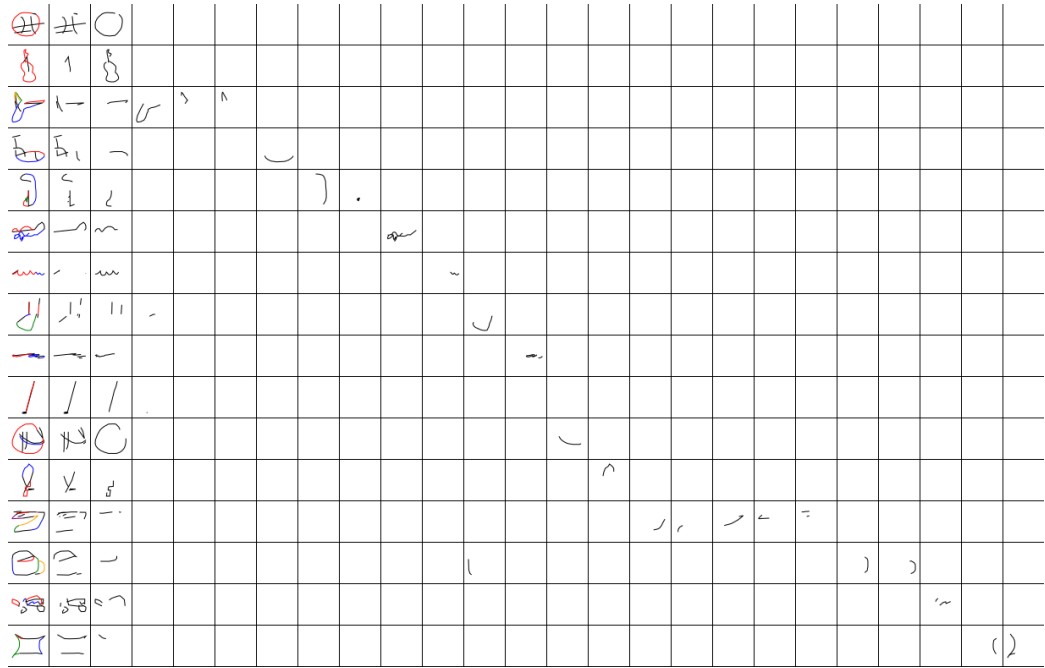

Figure 24: Snowman class stroke clustering

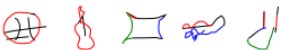

Figure 25: Fully composed images with coloured cluster assignments

