# OpenReview forum: "SketchEmbedNet: Learning Novel Concepts by Imitating Drawings"
_ICLR.cc/2021/Conference — Reject_

### Official Review · AnonReviewer2 · 2020-10-25
**contributions are not very clear**

**Rating:** 6
**Confidence:** 3

**Review:**

This paper proposes a generalized sketch drawing model named SketchEmbedNet for producing sketches and visual summaries of open-domain natural images. The idea is interesting and the experimental results show SketchEmbedNet is able to do not only few-shot classification but also one-shot generation.

Overall, I vote for rejecting. In my opinion, the main contributions of this paper are not very clear. The introduced model, SketchEmbedNet has limited novelty on neither the methodology nor the network structure. As stated in the title and introduction, the authors aim to capture and generalize the compositional information from sketch/natural images. Section 4 reports the latent variables organization performance, which is directly related to the key motivation I believe. But the authors only compared SketchEmbedNet with VAE, which is not enough to demonstrate their advantages. Moreover, it is not clear why few-shot classification and one-shot generation performance in Section 5 and 6 support their main idea. Thus, this paper needs further improvements.

Detailed comments:
(1)	In the first paragraph of section 2, the authors claimed that the CNN embeddings must preserve a compositional understanding of the input sketches/images to improve the performance in their pix2seq task. So how did you preserve the information? Many sketch synthesis methods, such as [8] in the reference, can reconstruct the sketch with a sketch image input. Do these methods preserve the ``compositional understanding’’?
(2)	Still in the same paragraph, I’m pretty sure a vanilla auto-encoder with a CNN encoder containing average pooling layers can reconstruct a six-legs-turtle well as input. Thus, the information about the positions and the turtle legs number must be transported to decoder by the latent embeddings. So why you confirm that a regular CNN embedding cannot preserve that information after an average pooling layer?
(3)	The same CNN encoder is used for both natural images and sketches. As these two types of images are with totally different patterns, many recent studies, such as reference [55], used a two-branched encoder for natural images and sketches, respectively. In this paper, the CNN backbone is a 4-layer CNN or a ResNet12, which are very basic structures. Is it able to extract the image features well?
(4)	Figure 5 shows SketchEmbedNet outperforms VAE on latent space organization. In my opinion, the well-organized latent space of SketchEmbedNet is mainly due to getting rid of the KL loss term, which drives the latent distribution to be a uniform distribution. I would like to see the comparison with sketch-pix2seq, which is the reference [8], as both SketchEmbedNet and sketch-pix2seq do not use KL term in training.
(5)	As this paper focuses on sketch drawing, why there is no comparison between SketchEmbedNet and other sketch generative models, such as [8, 11, 15, 26] in the references?




After reading the response from the authors, we raise our score by +1.

---

> ### Author Response · Authors · 2020-11-18
> **Response to reviewer #2**
>
> We thank reviewer 2 for their helpful review and suggestions.
>
> We have had some difficulty understanding the review. Some of the comments and citations do not appear to match the paper we submitted. We discuss compositional information in section 6, and few-show classification and one-shot generation in Sections 5 and 7 respectively. We have not used any bracket-style citations and citations 8, 11, 15, 26 do not all refer to sketching models, additionally, we do not have 55 citations in this work. However, we believe we are able to infer that [8] refers to sketch-pix2seq by Chen et al. (2017) and [55] refers to “Learning to sketch with shortcut cycle consistency” by Song et al. (2018) and will respond with this in mind. We are unable to identify references 11, 15 and 26.
>
> A general comment; our submission is exploring a novel setting where we train an image-to-sketch model on a very large number of classes to learn a generalized drawer and embedding function for downstream tasks. We demonstrate that this challenging task learns a unique embedding that captures salient and compositional visual information that we support by our experiments in Sections 5, 6 and 7. While it is a model that can sketch, we cannot directly compare to drawing models focused on producing drawings of few or singular classes and/or do not sketch image inputs.
>
> > How did you preserve the information [of compositional input images]? Many sketch synthesis methods, such as [8] in the reference, can reconstruct the sketch with a sketch image input. Do these methods preserve the ``compositional understanding’’?
>
> Like some other sketching methods, we motivate the preservation of salient and informative features through the use of our sequential RNN decoder. However, of these methods, only Sketch-pix2seq could potentially learn this information for rasterized image data and unlike our approach is limited in class scope and not intended for use on downstream tasks.
>
> > A vanilla auto-encoder with a CNN encoder containing average pooling layers can reconstruct a six-legs-turtle well as input. Thus, the information about the positions and the turtle legs number must be transported to decoder by the latent embeddings. So why you confirm that a regular CNN embedding cannot preserve that information after an average pooling layer?
>
> We present this observation from existing literature on CNN encoders with only pixel-reconstruction objectives. We have also rewritten this section to better convey our observation that it is the explicit, sequential generation of components by an RNN that leads to this awareness.
>
> > In this paper, the CNN backbone is a 4-layer CNN or a ResNet12, which are very basic structures. Is it able to extract the image features well?
>
> Yes we do believe our CNN backbone is capable of extracting meaningful image features. If our encoder has not extracted any salient information, we would not expect our results especially in few-shot classification.
>
> > In my opinion, the well-organized latent space of SketchEmbedNet is mainly due to getting rid of the KL loss term, which drives the latent distribution to be a uniform distribution. I would like to see the comparison with sketch-pix2seq, which is the reference [8], as both SketchEmbedNet and sketch-pix2seq do not use KL term in training.
>
> We do not explore latent space class disentanglement in the same way as Sketch-pix2seq. Our results in Section 6 are focused on understanding component spatial relationships in image space, and conceptual composition. The image examples embedded in Figure 5-A and 5-B are not classes that have been encountered by SketchEmbedNet but are synthetic examples created to demonstrate this understanding. Furthermore, this is only possible given our ability to encode general image inputs.
>
> However, to address your concern regarding the impact of  the KL term, we have repeated the experiments in Section 6 both using a standard autoencoder (a VAE without KL regularization) and a SketchEmbedNet trained with the KL objective. We do find that performance on the linear classification and regression tasks improve as we reduce KL regularization and have removed the experiment from our paper. However, SketchEmbeddings outperform all autoencoder based methods in our other experiments in section 6. We include the full set of results considering KL regularization in Appendix sections I and L.
>
> > As this paper focuses on sketch drawing, why there is no comparison between SketchEmbedNet and other sketch generative models, such as [8, 11, 15, 26] in the references?
>
> This is addressed in the general comments. While our model is a sketching model, we learn a mapping for different domains (general image inputs to sketch outputs) and class breadths that does not meaningfully compare to existing sketch models.

---

### Official Review · AnonReviewer1 · 2020-10-28
**Nice approach to learning image embeddings based on vector sketches but requires some additional clarification and discussion**

**Rating:** 6
**Confidence:** 4

**Review:**

### Summary

This paper proposes learning embeddings for sketch or natural images by training a network that takes in a raster image and outputs and collection of sketch strokes. The architecture consists of a standard CNN encoder followed by an RNN decoder. The authors evaluate their learned embeddings on few-shot classification tasks and explore the the quality of the latent space. They demonstrate that they outperform unsupervised few-shot classification approaches and seem to obtain a latent space that is more aware of long-range structure than those from methods that operate purely in raster space.

---

### Explanation of rating

Interpreting and synthesizing sketches in a deep learning context is an promising research direction. While the idea of focusing on of sketch-aware embeddings of images is an interesting one, the main technical contribution simply involves taking a standard convolutional encoder with a recurrent decoder, which has already been used for sketch generation (sketch-rnn). In addition to this, some of the claims made in the paper require some clarification or additional experimentation, as I explain below. Thus, I believe that some additions and changes must be made for the paper to be accepted.

---

### Pros

- I like the idea of using the vector structure of sketches to gain more insight into image content.
- It is difficult to design deep networks that are able to operate on non-raster data. In some sense, this approach sidesteps this issue by allowing the relationship between raster and vector to be learned.
- The experiments in Section 6 confirm the idea that the embeddings are aware of this high-level long-range relationships that aren't so obvious on the pixel level.

---

### Cons and questions

- I am not sure I follow the intuition behind why the proposed model achieves better semantic awareness than a convolutional VAE. The authors use the example of a six legged turtle and state that the VAE would only retain information about a single "leg-like feature but not how many legs are present." How is having an RNN stroke-based decoder different from a standard convolutional decoder in this sense? In both cases, the final reconstruction must contain the original number of legs, and so the latent vector is encouraged to retain this information.
- The performance on natural images in Figure 4, especially on unseen classes, is not great. I would be interested to see the nearest neighbor images in the training set for the examples shown. Even in the case of unseen classes, the resulting sketches look like they might match a similar image from the training set.
- The authors claim that balancing the stroke and and pixel losses via a curriculum mirrors how humans learn to draw ("paint-by-numbers"). However, I'm not sure how some of the experiments fit into this methodology. In particular, most of the experiments are done on datasets that have ground truth sketch strokes but not their ordering (e.g., SVG files). In this case, it seems like imposing an order on the strokes (and asking the decoder to replicate it) is a counterintuitive constraint for the model. On the other hand, in the natural image experiments, the pixel loss cannot be used at all. Some discussion of the consequences there would be interesting.
- I'm not sure that it is fair to compare to fully unsupervised few-shot classification methods. While the proposed method indeed does not use class labels, the ground truth stroke information may provide considerably more information than just the raster image data. Perhaps it would help to have a baseline without stroke loss (i.e., $\alpha=1$). Even the ablation study in Table 3 does not include this case.
- How sensitive is the approach to quality of stroke decomposition? For instance, what happens if you subdivide each stroke in the ground truth SVGs?

---

The authors have addressed many of my concerns in the rebuttal, and so I am increasing my rating.

---

> ### Author Response · Authors · 2020-11-18
> **Response to reviewer #1**
>
> We thank Reviewer 1 for their thorough and insightful review.
>
> > While the idea of focusing on of sketch-aware embeddings of images is an interesting one, the main technical contribution simply involves taking a standard convolutional encoder with a recurrent decoder, which has already been used for sketch generation (sketch-rnn).
>
> We aimed to build upon SketchRNN; in contrast to that approach, our model is class agnostic and not constrained to a single or a few input classes. The combination of this with a convolutional encoder is what allows us to demonstrate the salient and compositional nature of SketchEmbeddings and broad effectiveness for classical image-domain tasks.
>
> > The authors use the example of a six legged turtle and state that the VAE would only retain information about a single "leg-like feature but not how many legs are present." How is having an RNN stroke-based decoder different from a standard convolutional decoder in this sense?
>
> The difference is that an RNN decoder must sequentially and explicitly generate each element, which learns a greater component level awareness. A raster based reconstruction does not have this componential understanding and only matches pixels. We have updated this passage in our manuscript to better communicate the idea.
>
> > The performance on natural images in Figure 4, especially on unseen classes, is not great. I would be interested to see the nearest neighbor images in the training set for the examples shown. Even in the case of unseen classes, the resulting sketches look like they might match a similar image from the training set.
>
> While better drawings are desirable, it is not our primary focus to produce the most high-fidelity sketches. We find that our approach is capable of producing sketches that capture the major shapes of subjects in the image and that the distilled embedding of this information is sufficient to drive competitive performance in few-shot classification. The key demonstration that the model is not simply indexing into the training images comes from the few-shot classification results, which show that the embeddings are capable of discriminating between examples of completely novel classes.
>
> > The authors claim that balancing the stroke and and pixel losses via a curriculum mirrors how humans learn to draw ("paint-by-numbers"). … In particular, most of the experiments are done on datasets that have ground truth sketch strokes but not their ordering (e.g., SVG files).
> > Perhaps it would help to have a baseline without stroke loss (i.e., α=1). Even the ablation study in Table 3 does not include this case.
>
> The SVG data from datasets like Sketchy include timestamps over all paths of how the sketch was initially drawn. We preserve this sequence by sampling along those paths to obtain our training data. We have added a comment about this in the paper, to clarify how we use the SVG data. We have attempted to train models using only the pixel loss (α=1) but the model fails to converge thus our motivation for the paint-by-numbers approach.
>
> > On the other hand, in the natural image experiments, the pixel loss cannot be used at all. Some discussion of the consequences there would be interesting.
>
> This is not quite correct -- the pixel loss is used for natural image training. In our natural image training data from Sketchy we have stroke information for each training example. We convert this to an image via our rasterization and compare it to the drawing produced by our model, in order to compute pixel loss.
>
> > I'm not sure that it is fair to compare to fully unsupervised few-shot classification methods. While the proposed method indeed does not use class labels, the ground truth stroke information may provide considerably more information than just the raster image data.
>
> We have added a note in Table 1 and 2 to clarify this. While stroke data arguably contains more information we do not see it as a problem as we demonstrate that the learned information is informative and transferable across data domains. While Omniglot contains some stroke distribution, learning only to draw Quickdraw’s distribution addresses the task; while mini-ImageNet has no stroke distribution, learning to summarize natural images in the style of Sketchy is still informative for the former.
>
> >How sensitive is the approach to quality of stroke decomposition? For instance, what happens if you subdivide each stroke in the ground truth SVGs?
>
> Subdividing SVGs does not have a significant impact on our model. Our data is not in the form of a set of paths but a sequence of points. Subdividing a path in SVG data would equate to lifting and then immediately replacing the pen to the drawing canvas.

---

> > ### Comment · AnonReviewer1 · 2020-11-21
> > **Response**
> >
> > Thank you for the response. You have clarified some of my misunderstandings---in particular I had initially thought that you had trained an instance of your model from scratch on mini-ImageNet, hence my comments regarding pixel loss---and I appreciate the changes you made to the text.
> >
> > Some of my concerns still remain. I don't think the fact that your models transfer to other datasets counteracts the pac that your initial models are **not** trained in an unsupervised fashion, at least not in the same way as the works you compare to. Additionally, I do think that stability to stroke ordering or subdivision is obvious but rather something that would benefit from an ablation study; I see your point that, in theory, subdividing, should not change the representation, but it would be nice to validate tis in practice. Similarly, what happens if an identical set of strokes is drawn in a different order? Is the output of the model still the same?
> >
> > Having said that, I have upgraded my score thanks to the updates and clarifications.

---

> > > ### Author Response · Authors · 2020-11-24
> > > **Further response to reviewer #1**
> > >
> > > We thank Reviewer 1 for their consideration of our response and further comments.
> > >
> > > > I don't think the fact that your models transfer to other datasets counteracts the pac that your initial models are not trained in an unsupervised fashion, at least not in the same way as the works you compare to.
> > >
> > > We agree it is not entirely _fair_ to compare a model using stroke supervision with one that only uses an image input but it is also flawed to compare our methods to a supervised model given object classes during training. Our aim has been to show that drawing is a very useful auxiliary task for learning visual information and one way we demonstrate this is through effectively recognizing novel classes in an entirely different domain.
> > >
> > > Additionally, training with sequential data and training with only image data have been compared since the proposal of the Omniglot dataset by creators Lake et al. in their initial (2015) and follow-up works (2019).
> > >
> > > Brenden M. Lake, Ruslan Salakhutdinov, and Joshua B. Tenenbaum. Human-level concept learning through probabilistic program induction. *Science*, 350(6266):1332–1338, 2015. doi: 10.1126/ science.aab3050.
> > >
> > > Brenden M Lake, Ruslan Salakhutdinov, and Joshua B Tenenbaum. The omniglot challenge: a 3-year progress report. *Current Opinion in Behavioral Sciences*, 29:97–104, Oct 2019.
> > >
> > > > In theory, subdividing, should not change the representation, but it would be nice to validate tis in practice. Similarly, what happens if an identical set of strokes is drawn in a different order? Is the output of the model still the same?
> > >
> > > Thank you for the suggestion but we would like to make some clarifications before we consider this ablation study.
> > >
> > > Our model only uses stroke information during the batched training task in Section 4. Any embedding used in Sections 5, 6, 7 is produced by the CNN encoder from Section 4 which only takes in a pixel image. This means that evaluating the impact of subdivision and shuffling on our latent space would entail shuffling/slicing the Quickdraw/Sketchy training data from Section 4, repeating those experiments and then re-evaluating Sections 5, 6 and 7.
> > >
> > > Furthermore, our Section 4 training data already contains varying stroke trajectories and subdivisions (which you have also noticed is not hugely significant given our data representation). Quickdraw examples contain many images that are drawn with different or divided trajectories, for example, circles drawn clockwise or counterclockwise. Sketchy further extends this by having up to 10 unique stroke “labels” for each natural-image input as per the original dataset. We could reshuffle or slice these examples but this would only at best shift our training distribution in Section 4. As such, we are hesitant to anticipate any significant impact on our learned embedding function from Section 4 and its evaluations Sections 5, 6 and 7.

---

### Official Review · AnonReviewer4 · 2020-10-29
**Initial review**

**Rating:** 4
**Confidence:** 3

**Review:**

A brief summary:
This paper shows that the model trained to restore sequential data from images in a supervised manner tends to capture more informative latent representations of the data.

Strengths:
+ Demonstrates that training a decoder model to reconstruct sequential sketches leads for an encoder to better represent the input image.
+ Achieves the SOTA result in Omniglot recognition task, compared against existing unsupervised methods.

Weaknesses:
- Confuses readers with the ambiguous claim. The embedding function is said to produce `salient and compositional features' (p1), but no evaluations on compositions were included in the main paper.
- Does not include any ablation studies to show the effectiveness of each components of the proposed model.
- Does not include thorough explanations or analysis on each set of experiments.

Initial recommendation:
Borderline reject.

Reasons:
- While this paper provides a lot of experimental results, which I very much appreciate, I still found most of them quite irrelevant to support the main claim by the authors, which discusses compositional embeddings. The main contribution of this paper, I believe, is mainly the idea to utilize the sequential sketch data during the supervised training time. This needs to be clearly stated, particularly inside the tables, as most baselines only use sketch images, not their sequential data.
That being said, this trick is shown to work well, and if these significant improvement on Omniglot is further verified, this trick will be found useful by the community of researchers who work with sequential data, such as sketches or handwriting.
- However, I found most of the implementation details quite unclear, and experimental results were often misleading. For instance, the results from Table 4, if we look closely, the authors' results use ResNet12 trained with Sketchy data, which quite differs from other results in the list.

Feedbacks:
- Authors briefly mention in p.4 that a KL term did negatively effect the performance, but no detailed explanations/experiments were given.
- What happens when you provide the class labels and train the model with classification errors as well? Will it improve the test result? If so, how much?
- May have been much more interesting if this paper explores the unsupervised disentanglement in latent space, to support their claims for what they note as structured embeddings. Will other latent disentanglement methods (e.g. beta-, factor-, VQ-VAE, etc.) lead to better representations?
- Conv-VAE alone may not be the best baseline.

---

> ### Author Response · Authors · 2020-11-18
> **Response to reviewer #4, part 1**
>
> We thank Reviewer 4 for their detailed insights and helpful initial review of our work.
>
> > The embedding function is said to produce `salient and compositional features' (p1), but no evaluations on compositions were included in the main paper.
> > While this paper provides a lot of experimental results, which I very much appreciate, I still found most of them quite irrelevant to support the main claim by the authors, which discusses compositional embeddings.
>
> Our main claims of learning ‘salient and compositional features’ are supported by our experimental results in Sections 5, 6 and 7. We demonstrate salience in Sections 5 and 7 by showing that sketching is an informative task that produces embeddings that lead to strong few-shot classification results, and by generating recognizable examples from them. We showcase an understanding of component composition in image space and conceptual composition in our experiments in section 6. The introduction and experimental sections have been updated with additional discussion to better connect our claims with these results.
>
> > Does not include any ablation studies to show the effectiveness of each components of the proposed model.
>
> We ablate a key term, the pixel loss, by varying the αmax coefficient. We also investigated ablating the stroke loss, by setting α=1, but this fails to converge motivating our curriculum learning schedule. Ablating the pen loss would mean our output pen states would be generally unlearned. Other modifications, such as altering the encoder-decoder architecture, either changes our problem formulation entirely or makes our model a Conv-VAE.
>
> > [The main contribution of this paper is] to utilize the sequential sketch data during the supervised training time. This needs to be clearly stated, particularly inside the tables, as most baselines only use sketch images, not their sequential data.
>
> We view our contribution as showing that learning to draw via the sequential sketch data is a useful task for learning visual concepts. We do agree that this additional information used during training is not used by the methods we are comparing to in the few-shot classification results, and have made sure to highlight this in the revised version (with a note next to the tables). Nonetheless we want to emphasize that this additional information is only present during training time yet transfers well for other datasets (Quickdraw -> Omniglot, Sketchy -> mini-ImageNet) with only image inputs during test time. We think it is an interesting contribution to show that adding this auxiliary sketch information during training, on different datasets and different classes, can help few-shot classification.
>
> > For instance, the results from Table 4, if we look closely, the authors' results use ResNet12 trained with Sketchy data, which quite differs from other results in the list.
>
> All results in Table 4 use the same architecture. If you are referring to Table 2, we have provided additional comparisons with a Conv-VAE using a ResNet12 backbone as well as SketchEmbedNet using the Conv4 architecture. ResNet12 was selected to accommodate the increased challenge in sketching natural images as the model must handle color and identify the subject. While our Conv4 results are comparable or weaker than existing methods we believe that even without state-of-the-art, our use of sketch information for visual tasks is an interesting and novel contribution.

---

> > ### Author Response · Authors · 2020-11-18
> > **Response to reviewer #4, part 2**
> >
> > > Authors briefly mention in p.4 that a KL term did negatively effect the performance, but no detailed explanations/experiments were given.
> >
> > Including a KL loss saw a minor reduction in quantitative performance in sections 5 and 7 and we believe it offers little benefit. While the regularizer is useful to prevent memorization and reproducing the exact image input, we are performing image to sketch transfer so this issue is less of a concern. We expanded the discussion of this in the paper, and have included this as an additional entry in the table in Appendix I.
> >
> > > What happens when you provide the class labels and train the model with classification errors as well? Will it improve the test result? If so, how much?
> >
> > We agree that this is interesting, as one would expect using labels would learn a more discriminative embedding.We ran this experiment and found that learning with class-labels during the supervised training stage worsens classification accuracy -- we have included these results in Appendix L. While sketching is informative across classes and data distributions, explicit classification tasks do not seem to be.
> >
> > > May have been much more interesting if this paper explores the unsupervised disentanglement in latent space, to support their claims for what they note as structured embeddings. Will other latent disentanglement methods (e.g. beta-, factor-, VQ-VAE, etc.) lead to better representations?
> > > Conv-VAE alone may not be the best baseline.
> >
> > We agree that it would be useful to extend our baselines to include other embedding approaches, particularly ones aimed at learning disentangled representations. We have investigated a beta-VAE for this purpose but the results appear similar to the Conv-VAE, which we have included in Appendix sections I and L. We intend to add another disentanglement approach, such as factor-VAE, in a further revision of the paper.

---

### Official Review · AnonReviewer3 · 2020-11-06
**High quality submission with impressive and promising results**

**Rating:** 9
**Confidence:** 4

**Review:**

Authors investigate the possibility to learn a generalized embedding function that captures salient and compositional features of sketches by directly imitating human sketches. The manuscript is written clearly and concisely. Methods have been presented with enough detail and seem accurate. Particularly, the results from the Quickdraw and Omniglot datasets showing generated sketches are rather impressive, and the ones for the natural images seem promising. Overall, I very much enjoyed reading the paper and suggest it for publication without any major changes.

In my view the results presented in Figure 5, and especially 5C, are the most impressive and interesting ones. These results deserve more space in the manuscript. I was curious to know whether there were also many unsuccessful conceptual composition examples. Are the examples shown in Figure 5C the best ones, or are they representative of performance in general? Does this approach also work with natural images to any extent? Could the authors elaborate on why or why not this may be the case?

---

> ### Author Response · Authors · 2020-11-18
> **Response to reviewer #3**
>
> We’d like to thank Reviewer 3 for their comments and helpful suggestions.
>
> > I was curious to know whether there were also many unsuccessful conceptual composition examples. Are the examples shown in Figure 5C the best ones, or are they representative of performance in general?
>
> We have added additional conceptual compositionality examples to Appendix K and aim to expand the experiment with more class types. The challenge has not been getting SketchEmbedNet to produce good results, but rather, in constructing intuitive tasks for this phenomenon.
>
> >Does this approach also work with natural images to any extent? Could the authors elaborate on why or why not this may be the case?
>
> We haven’t tested this yet as finding suitable classes and perspectives is a challenge and would like to focus on improving decoding quality. However, as we decode natural images and sketches into the same space, we are optimistic that the SketchEmbeddings would exhibit the same properties when composed!

---

### Author Response · Authors · 2020-11-18
**General remarks**

We’d like to thank all the reviewers for their input and thoughtful reviews.

In addition to the changes we’ve made to our paper in response to each review, we’ve updated section 6 of the manuscript. Figure 5-A and 5-B have been updated with new colors, and we have changed the image generation method to prevent overly thick strokes as images are resized. We have also updated the writing in the discussion portion.

We have also included our appendices in the main PDF rather than only in the supplementary materials file.

---

### Decision · Program_Chairs · 2021-01-07
**Final Decision**

**Decision:**

Reject

**Comment:**


Description:
The paper presents a generative model, SketchEmbedNet, for class-agnostic generation of sketch drawings from images. They leverage sequential data in hand-drawn sketches. Results shows this outperforms STOA on few-shot classification tasks, and the model can generate sketches from new classes after one shot.

Strengths:
- Detailed, technically sound, presentation
- Shows that enforcing the decoder to output sequential data leads to a more informative internal representation, and thus generate better quality sketches
- Improves over unsupervised STOA methods

Weaknesses:
- Experiments are done against methods that do not use the sequential aspect of sketches. Because ground-truth in this case contains much more data, comparison is not quite fair.
- Will have been useful to see results against a baseline that uses it.
- Quality of sketches generated from natural images is low